# Satellite cell-derived TRIM28 is pivotal for mechanical load- and injury-induced myogenesis

Kuan-Hung Lin[1,2,3,7], Jamie E Hibbert[1,2,7], Corey GK Flynn[1,2], Jake L Lemens[1,2], Melissa M Torbey [1,2], Nathaniel D Steinert[1,2], Philip M Flejsierowicz[1,2], Kiley M Melka[1,2], Garrison T Lindley[1,2], Marcos Lares[4], Vijayasaradhi Setaluri[4], Amy J Wagers [3,5,6] & Troy A Hornberger [1,2 ✉]

## Abstract

Satellite cells are skeletal muscle stem cells that contribute to postnatal muscle growth, and they endow skeletal muscle with the ability to regenerate after a severe injury. Here we discover that this myogenic potential of satellite cells requires a protein called tripartite motif-containing 28 (TRIM28). Interestingly, different from the role reported in a previous study based on C2C12 myoblasts, multiple lines of both in vitro and in vivo evidence reveal that the myogenic function of TRIM28 is not dependent on changes in the phosphorylation of its serine 473 residue. Moreover, the functions of TRIM28 are not mediated through the regulation of satellite cell proliferation or differentiation. Instead, our findings indicate that TRIM28 regulates the ability of satellite cells to progress through the process of fusion. Specifically, we discover that TRIM28 controls the expression of a fusogenic protein called myomixer and concomitant fusion pore formation. Collectively, the outcomes of this study expose the framework of a novel regulatory pathway that is essential for myogenesis.

**Keywords** Skeletal Muscle; Hypertrophy; Cell Fusion; Regeneration
**Subject Categories** Musculoskeletal System; Stem Cells & Regenerative Medicine

## Introduction

As the largest tissue in the body, skeletal muscle comprises ~40% of total mass and plays an essential role in breathing, mobility, whole-body metabolism, and maintaining a high quality of life (Izumiya et al, 2008; Seguin and Nelson, 2003). Notably, adults will lose ~35–40% of their muscle mass between the ages of 25 and 80 years (Janssen et al, 2000). This gradual loss of muscle is not only associated with decreased independence, disability, and mortality but also an estimated 40 billion dollars in annual healthcare costs in the United States alone (Goates et al, 2019; Pahor and Kritchevsky, 1998). Therefore, developing therapies that can maintain or restore muscle mass is of great clinical and fiscal significance. However, to develop such therapies, a comprehensive understanding of the molecular mechanisms that regulate this system is needed.

It is well established that skeletal muscles possess a remarkable ability to adapt their mass in response to conditions such as an increase in mechanical loading, and they are also capable of regenerating after an injury (Carlson, 1973; Zhu et al, 2021). At the cellular level, the mechanical load-induced increase in muscle mass is thought to be driven by an increase in the size of the pre-existing myofibers, whereas regeneration is driven by the formation of new myofibers (Jorgenson et al, 2020; Tedesco et al, 2010). Importantly, a class of skeletal muscle stem cells called satellite cells have been implicated in the regulation of both processes (Collins et al, 2005; Murach et al, 2018). Specifically, in mature skeletal muscle, satellite cells are quiescent but they can be activated by stimuli such as an increase in mechanical loading or injury (Eliazer et al, 2019; Murach et al, 2021). Once activated, satellite cells will express a critical myogenic transcription factor called the myoblast determination protein (MYOD), and then enter the cell cycle to expand their population (Cooper et al, 1999; Yin et al, 2013). Most of the proliferated satellite cells will then differentiate into myoblasts and contribute to myogenesis by undergoing heterotypic fusion (the fusion between a myoblast and myofiber) or homotypic fusion (the fusion between myoblasts) which will give rise to the accretion of new myonuclei in pre-existing myofibers or the formation of new myofibers, respectively (Hindi et al, 2013; Masschelein et al, 2020; Sampath et al, 2018).

Satellite cells, like all other cells in the body, express tripartite motif-containing 28 (TRIM28) (Kim et al, 1996; Petrany et al, 2020). TRIM28 functions as a global genome transcription intermediary factor and it has been reported to be involved in a variety of biological events, including embryonic and stem cell development, cell cycle regulation, tumorigenesis, repression of endogenous retroviruses, and, of particular interest for this study, myogenesis (Cammas et al, 2000; Kim et al, 2015; Lionnard et al,

[1]Department of Comparative Biosciences, University of Wisconsin - Madison, Madison, WI, USA. [2]School of Veterinary Medicine, University of Wisconsin - Madison, Madison, WI, USA. [3]Department of Stem Cell and Regenerative Biology, Harvard University, Cambridge, MA, USA. [4]Department of Dermatology, University of Wisconsin - Madison, Madison, WI, USA. [5]Harvard Stem Cell Institute, Cambridge, MA, USA. [6]Joslin Diabetes Center, Boston, MA, USA. [7]These authors contributed equally: Kuan-Hung Lin, Jamie E Hibbert. ✉E-mail: troy.hornberger@wisc.edu

2019; Singh et al, 2015). Specifically, a previous study in C2C12 myoblasts reported that the changes in TRIM28(S473) phosphorylation can act as a switch that regulates the ability of MYOD to drive myogenesis in vitro (Singh et al, 2015). We were intrigued by this possibility because recent phosphoproteomic analyses from our lab revealed that an increase in mechanical loading leads to a robust increase in TRIM28(S473) phosphorylation and therefore suggested that TRIM28 might serve as a critical regulator of myogenesis in vivo (Potts et al, 2017; Steinert et al, 2021). Thus, the overarching goal of this study was to address this possibility.

## Results

### Maximum intensity contractions induce TRIM28(S473) phosphorylation in satellite cells

To identify phosphorylation events that regulate skeletal muscle growth, our lab performed two different phosphoproteomic-based studies on mouse tibialis anterior (TA) muscles that had been subjected to mechanical loading via a bout of maximal-intensity contractions (MICs). In both studies, changes in TRIM28(S473) phosphorylation were identified as one of the most robustly regulated events (Potts et al, 2017; Steinert et al, 2021), and as shown in Fig. 1A,B, we extended those findings by showing that the induction of TRIM28(S473) phosphorylation could be detected in myonuclei as well as within the nuclei of a subset of the interstitial cells. Intrigued by this finding, and the report that TRIM28(S473) phosphorylation regulates myogenesis in vitro (Singh et al, 2015), we performed additional immunohistochemical analyses to determine whether MICs induce TRIM28(S473) phosphorylation within satellite cells. As shown in Fig. 1C,D, we found that MICs lead to a robust increase in the percentage of satellite cells that were positive for TRIM28(S473) phosphorylation and therefore suggested that changes in TRIM28(S473) phosphorylation might play a key role in regulating the myogenic effects of mechanical loading.

### Characterization of tamoxifen-inducible and satellite cell-specific TRIM28 knockout mice

To define the role of TRIM28 in myogenesis, we generated tamoxifen-inducible and satellite cell-specific *Trim28* knockout mice by crossing *Trim28*flox/flox mice with Pax7-iCre+/− mice (Cammas et al, 2000; Murphy et al, 2011). Offspring with a genotype of Pax7-iCre+/− : *Trim28*flox/flox were used for the knockout group (KO), and those with a genotype of Pax7-iCre−/− : *Trim28*flox/flox were used for the control group (WT). To establish the specificity and efficiency of the knockout, we collected plantaris muscles from the WT and KO mice 14 days after they had been treated with tamoxifen. As shown in Fig. 1E–G, immunohisto-chemical analysis revealed that >98% of the satellite cells in the plantaris muscles from the KO mice were TRIM28 negative, whereas the expression of TRIM28 remained readily detectable in Pax7 negative cells (Fig. 1G). Importantly, we also determined that the number of satellite cells per myofiber was not altered in the KO condition which indicates that the loss of TRIM28 in satellite cells did not impact satellite cell viability (Fig. 1H).

### The loss of TRIM28 does not alter the mechanical load-induced proliferation or differentiation of satellite cells

Having confirmed the specificity and efficiency of the KO mice, our next goal was to determine whether TRIM28 plays a role in mechanical load-induced myogenesis. To accomplish this, we subjected tamoxifen-treated WT and KO mice to a unilateral synergist ablation (SA) surgery in which the soleus and distal two-thirds of the gastrocnemius muscles were surgically removed in one hindlimb, while a sham (control) surgery was performed on the contralateral hindlimb. Plantaris muscles were collected at 4 days post SA and subjected to immunohistochemical analysis for Ki67 which serves as a marker of proliferating cells (Gerdes et al, 1983; Sun et al, 2017). As shown in Fig. 2A,B, we determined that SA led to a substantial increase in the number of Ki67 positive satellite cells, and this effect was not impacted by the loss of TRIM28. We also subjected the same samples to immunohistochemical analysis of myogenin (MYOG) which serves as a marker of satellite cells that have differentiated into myoblasts. As expected, the outcomes revealed that SA led to a significant increase in the number of MYOG-positive cells per myofiber, but again, this effect was not impacted by the loss of TRIM28 (Fig. 2C,D). Collectively, these results indicate that TRIM28 is not required for the mechanical load-induced proliferation or differentiation of satellite cells.

### The loss of TRIM28 in satellite cells attenuates mechanical load-induced myonuclear accretion but it does not alter the increase in muscle mass or myofiber size

Current models assert that when mechanical loads induce the activation of satellite cells, the satellite cells will proliferate, differentiate, and then engage in heterotypic fusion with the pre-existing myofibers (Murach et al, 2021; Petrella et al, 2008; Randrianarison-Huetz et al, 2018; Smith et al, 2001). As reported in Fig. 2, the loss of TRIM28 did not impact the mechanical load-induced activation of satellite cell proliferation or differentiation, however, these experiments did not address whether TRIM28 was required for the induction of heterotypic fusion/myonuclear accretion. Therefore, to address this, WT and KO mice were subjected to SA and treated with daily injections of 5-bromo-2'-deoxyuridine (BrdU) for 14 days. BrdU labels DNA that has undergone replication, and since myonuclei have been shown to replicate DNA at an extremely low rate (Borowik et al, 2023), the vast majority of BrdU-labeled myonuclei should be representative of satellite cells that have undergone proliferation, differentiation, and subsequent heterotypic fusion (Pullman and Yeoh, 1978; Smith et al, 2001). As illustrated in Fig. 3A,B, the outcomes of this experiment revealed that SA led to a substantial increase in the number of BrdU-positive myonuclei, and the magnitude of this effect was significantly lower in the muscles of KO mice. We also determined that SA led to an increase in the number of myonuclei per myofiber, and again, the magnitude of this effect was significantly lower in the muscles of the KO mice (Fig. 3C). Since the loss of TRIM28 did not appear to alter the mechanical load-induced activation of satellite cell proliferation or differentiation (Fig. 2A,B), the most likely explanation for these results was that the loss of TRIM28 inhibited the process of heterotypic fusion.

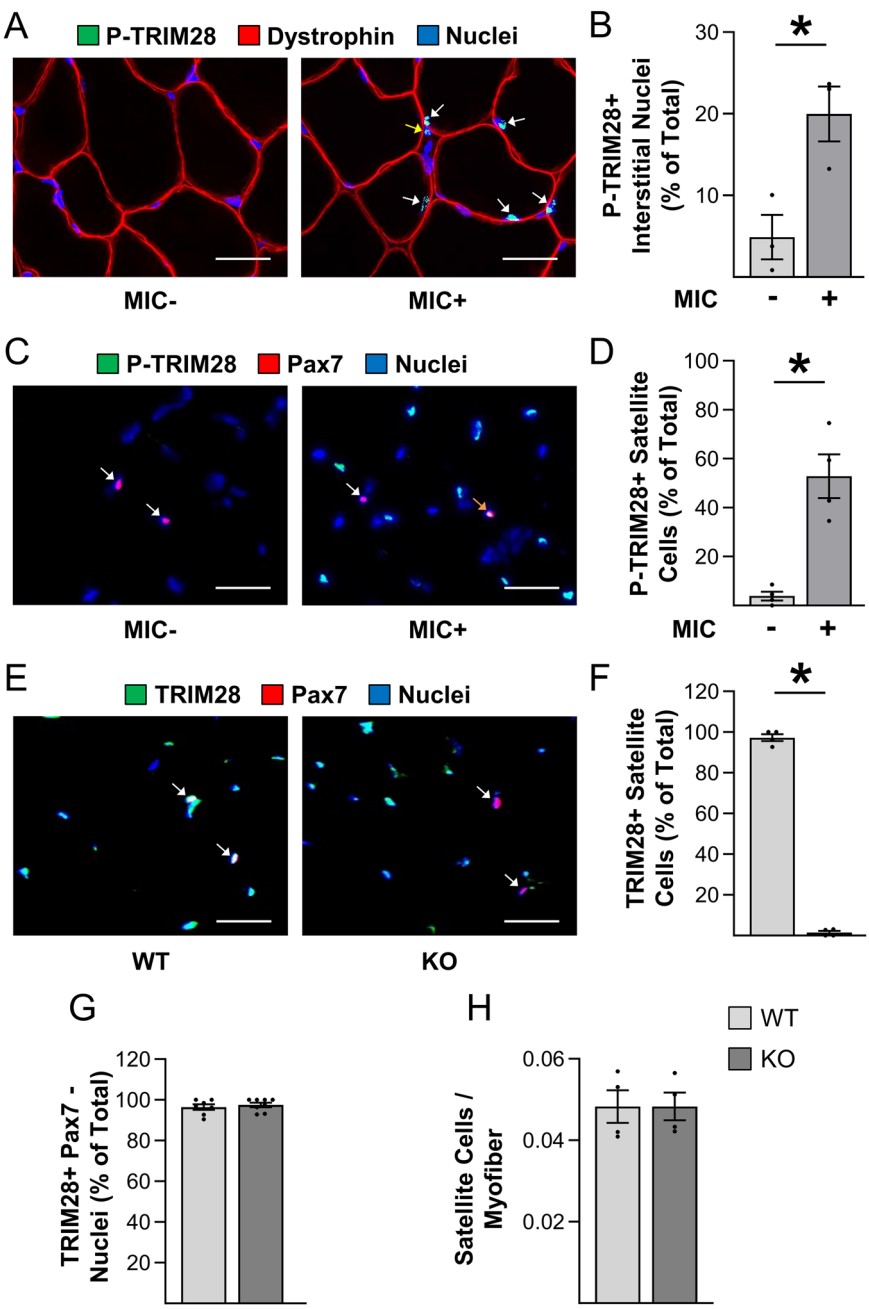

**Figure 1.  Maximal intensity contractions induce TRIM28(S473) phosphorylation in satellite cells.**

Tibialis anterior muscles of male wild-type (WT) mice were subjected to a bout of maximal-intensity contractions (MIC+) or the control condition (MIC−) and collected at 1 h post-treatment. (A) Mid-belly cross-sections of the tibialis anterior muscles were subjected to immunohistochemistry for phosphorylated (P) TRIM28(S473), dystrophin (to identify the outer boundary of the myofibers), and nuclei. White arrows indicate P-TRIM28 positive myonuclei and the yellow arrow indicates a P-TRIM28 positive interstitial nucleus. (B) Quantification of the % of P-TRIM28(S473) positive (+) interstitial nuclei from (A) (paired Student's t-test, $n = 3$/group, *$p = 0.0245$). (C) Mid-belly cross-sections of MIC− and MIC+ tibialis anterior muscles were subjected to immunohistochemistry for P-TRIM28(S473), Pax7 (to identify satellite cells), and nuclei. White arrows indicate P-TRIM28 negative satellite cells and the orange arrow indicates a P-TRIM28 positive satellite cell. (D) Images in (C) were used to quantify the percentage of satellite cells that were P-TRIM28(S473) positive (paired Student's t-test, $n = 4$/group, *$p = 0.0103$). (E) WT mice and tamoxifen-inducible satellite cell-specific TRIM28 knockout mice (KO) were treated with tamoxifen. At 14 days post tamoxifen, mid-belly cross-sections from the plantaris muscle were subjected to immunohistochemistry for TRIM28, Pax7, and nuclei. White arrows indicate satellite cells. (F–H) Quantification of the percentage of satellite cells that were TRIM28 positive (unpaired Student's t-test, $n = 4$/group, *$p < 0.0001$) (F) the percentage of Pax7 negative nuclei that were TRIM28 positive (unpaired Student's t-test, $n = 7$–8/group) (G), and the satellite cell to myofiber ratio (unpaired Student's t-test, $n = 4$/group) (H). All values are group means ± SEM. * indicates significant difference between groups, $P < 0.05$. Scale bars = 25 μm. Source data are available online for this figure.

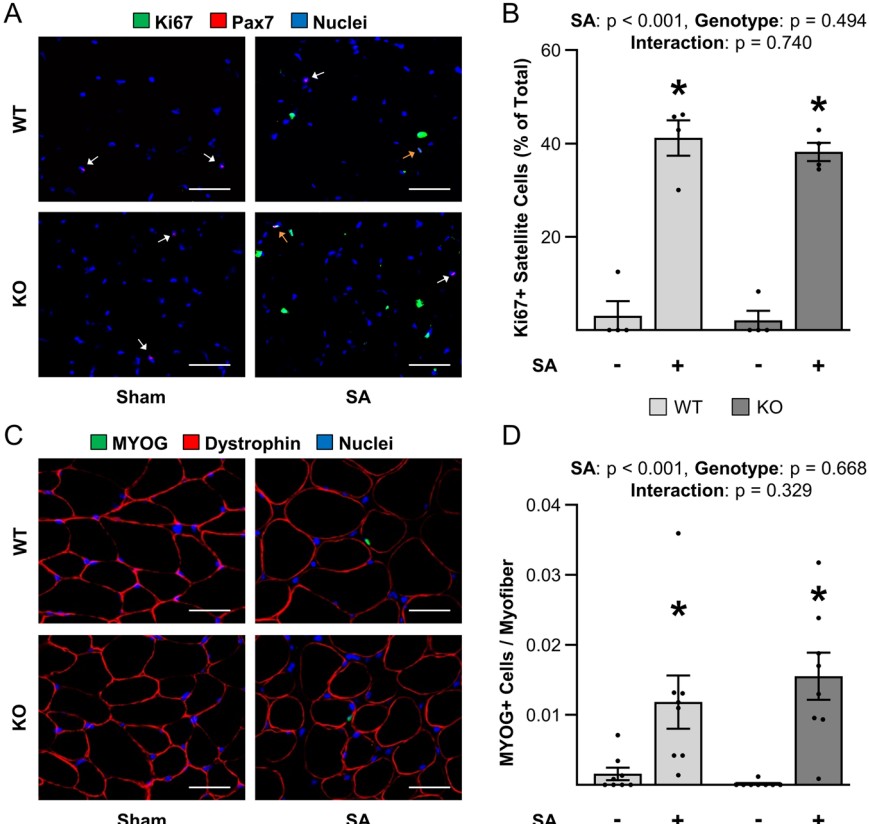

**Figure 2. The loss of TRIM28 does not alter the mechanical load-induced proliferation or differentiation of satellite cells.**

Wild-type (WT) mice and tamoxifen-inducible satellite cell-specific TRIM28 knockout mice (KO) were treated with tamoxifen. At 14 days post tamoxifen, the plantaris muscles of mice were subjected to a unilateral synergist ablation surgery (SA+), with the non-ablated limb serving as a sham control (SA−). The plantaris muscles were collected 4 days post-surgery and subjected to the following analyses. (A) Mid-belly cross-sections were subjected to immunohistochemistry for Pax7, Ki67 (a marker of proliferating cells), and nuclei. White arrows indicate Ki67 negative satellite cells and orange arrows indicate Ki67 positive satellite cells. (B) Quantification of the percentage of satellite cells in A that were Ki67 positive (two-way ANOVA, $n = 4$/group, *$p < 0.0001$). (C) Mid-belly cross-sections were subjected to immunohistochemistry for MYOG (a marker of differentiated satellite cells), dystrophin, and nuclei. (D) Quantification of the MYOG positive cell to myofiber ratio in (C) (two-way ANOVA, $n = 8$/group, *$p = 0.009$ or $0.0002$). Values are means ± SEM. * indicates a significant effect of SA within the given genotype, $p < 0.05$. Scale bars = 50 μm. Source data are available online for this figure.

However, alternative explanations such as TRIM28 being required for the post-differentiation survival of the satellite cells cannot be ruled out.

In conjunction with their role in myonuclear accretion, many studies have asserted that the satellite cell-dependent accretion of myonuclei is required for the mechanical load-induced growth of skeletal muscle (Goh and Millay, 2017; Goh et al, 2019; Guerci et al, 2012). As such, we reasoned that the reduced myonuclear accretion in the KO muscles would lead to impaired growth. However, as shown in Fig. 3D–F, we determined that the loss of TRIM28 did not alter the mechanical load-induced increase in mass or cross-sectional area (CSA) of the Type II myofibers (see Fig. EV1 for Type IIa, IIx, and IIb specific data). Nonetheless, many of the myofibers in the KO muscles that had been subjected to SA had an abnormal appearance of dystrophin, and this effect was not observed in the sham (control) muscles or in the muscles from WT mice that had been subjected to SA (Appendix Fig. S1). The abnormal appearance of dystrophin was similar to what has been described in various skeletal muscle myopathies (Selcen et al, 2004), and thus, although the satellite cell-specific loss of TRIM28 did not

inhibit mechanical load-induced growth at the time point that we studied, we cannot exclude the possibility that it may have led to functional deficits.

## The loss of TRIM28 in satellite cells leads to a fusion defect during injury-induced myogenesis

In addition to their role in the regulation of myonuclear accretion during the mechanical load-induced growth of skeletal muscle, numerous studies have also shown that satellite cells are required for the successful regeneration of skeletal muscle following an injury (Lepper et al, 2011; McCarthy et al, 2011; Sambasivan et al, 2011). Hence, we next sought to determine whether TRIM28 plays a role in injury-induced myogenesis/regeneration. To accomplish this, one TA muscle from each WT and KO mouse was injected with $BaCl_2$ to induce injury (Morton et al, 2019), whereas the contralateral TA was injected with phosphate-buffered saline (PBS) as a control condition. The number of MYOD-positive nuclei at 3 days post-injury (dpi) was then used to assess satellite cell activation and proliferation. As illustrated in Appendix Fig. S2A,B,

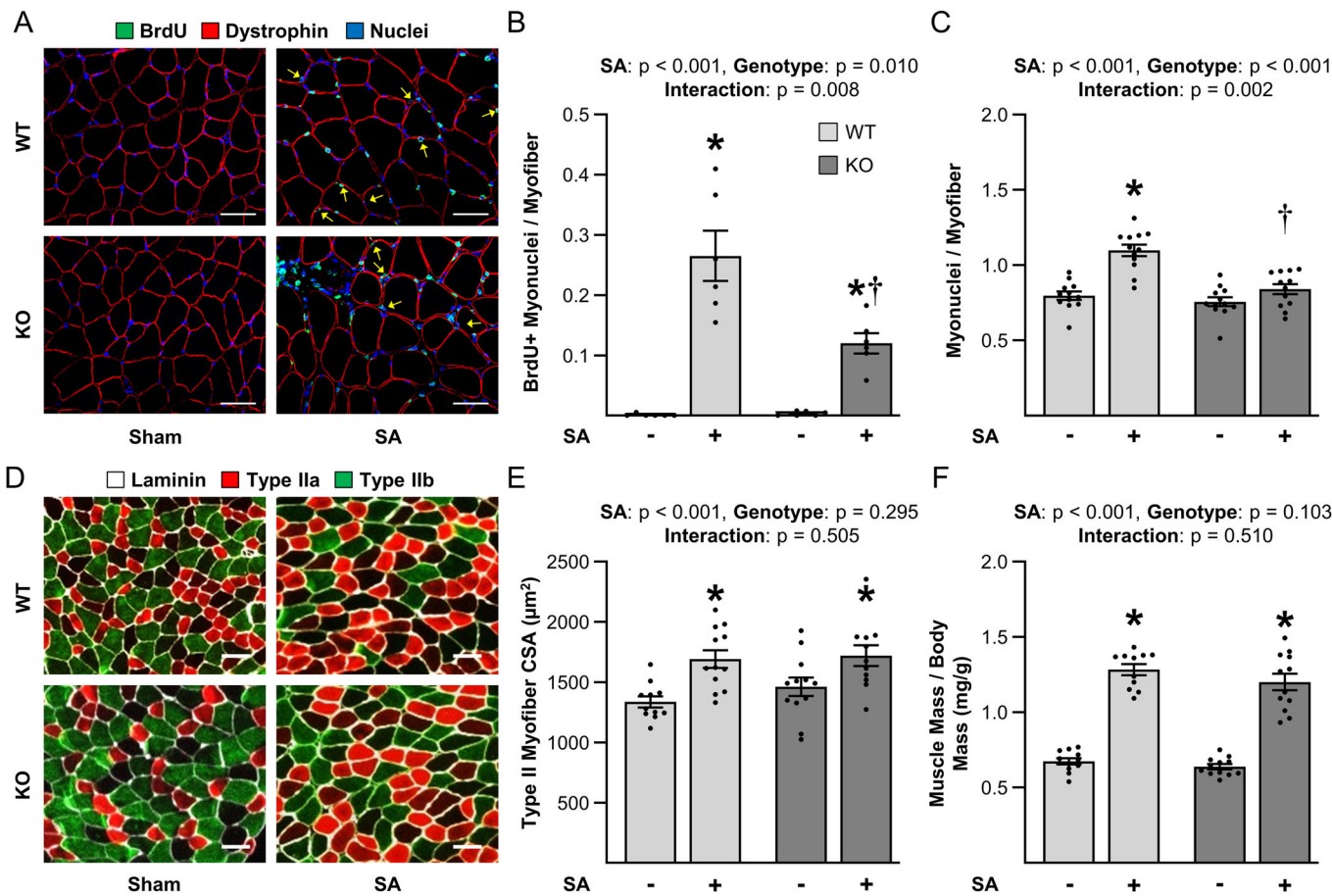

**Figure 3. The loss of TRIM28 in satellite cells attenuates mechanical load-induced myonuclear accretion but it does not alter the increase in muscle mass or myofiber size.**

Wild-type (WT) mice and tamoxifen-inducible satellite cell-specific TRIM28 knockout mice (KO) were treated with tamoxifen. At 14 days post tamoxifen, mice were subjected to unilateral synergist ablation surgery (SA+), with the non-ablated limb serving as a sham control (SA−). The mice were given daily intraperitoneal injections of BrdU for 14 days to label cells with replicated DNA and then the plantaris muscles were collected. (A) Mid-belly cross-sections were subjected to immunohistochemistry for BrdU, dystrophin, and nuclei. Yellow arrows indicate BrdU-positive myonuclei. (B) Quantification of the BrdU-positive myonuclei to myofiber ratio (two-way ANOVA, $n = 6$–9/group, *$p < 0.0001$ or $= 0.0015$, †$p = 0.0002$) and (C) the total myonuclei to myofiber ratio in A (two-way ANOVA, $n = 12$/group, * and †$p < 0.0001$). (D) Mid-belly cross-sections were subjected to immunohistochemistry for laminin and the identification of myofiber type. (E) The average CSA of the type IIa, IIx, and IIb myofibers were determined and reported as type II myofiber CSA (two-way ANOVA, $n = 11$–12/group, *$p = 0.0013$ or $0.0164$). (F) The muscle mass to body mass ratio (two-way ANOVA, $n = 11$–12/group, *$p < 0.0001$). Note: half of the mice in (E) and (F) were BrdU injected and the other half were not. Values are group means ± SEM.* indicates a significant effect of SA within the given genotype, † indicates a significant difference between the SA groups, $p < 0.05$. Scale bars = 50 μm. Source data are available online for this figure.

the injury led to a dramatic increase in the number of MYOD-positive nuclei, and this effect was not altered by the satellite cell-specific loss of TRIM28. Thus, consistent with the results from the SA model, it can be concluded that TRIM28 is not required for the activation and subsequent proliferation of satellite cells.

Next, we wanted to determine whether the satellite cell-specific loss of TRIM28 would impact the regeneration of the injured muscles. Hence, TA muscles were collected at 5, 7, 10, and 21 dpi, and subjected to immunohistochemistry for embryonic myosin heavy chain (eMHC) which serves as a marker of differentiated myoblasts and/or regenerating myofibers (Fig. 4A) (Guiraud et al, 2019; Schiaffino et al, 1986). In the TA muscles from WT mice, numerous small eMHC positive and centrally nucleated myoblasts/myofibers were detected at 5 and 7 dpi. By 10 dpi, the majority of the myofibers were eMHC negative and the overall CSA of the

myofibers had substantially increased. Finally, by 21 dpi, nearly all of the myofibers were eMHC negative, and aside from central nucleation, they were indistinguishable from the myofibers in the control muscles (Fig. 4A—top row). Similar to WT mice, the injured muscles of KO mice also contained numerous small eMHC positive and centrally nucleated myoblasts/myofibers at 5 and 7 dpi (Fig. 4A—bottom row). However, by 10 dpi, the appearance of the injured WT and KO muscles became markedly different. Specifically, unlike WT muscles, the vast majority of the myoblasts/myofibers in the injured KO muscles remained eMHC positive and had a very small CSA. By 21 dpi, the difference between the WT and KO muscles became even more extreme, with the previously injured region in the KO muscles showing a dense infiltration of nuclei that was largely devoid of myoblasts/myofibers.

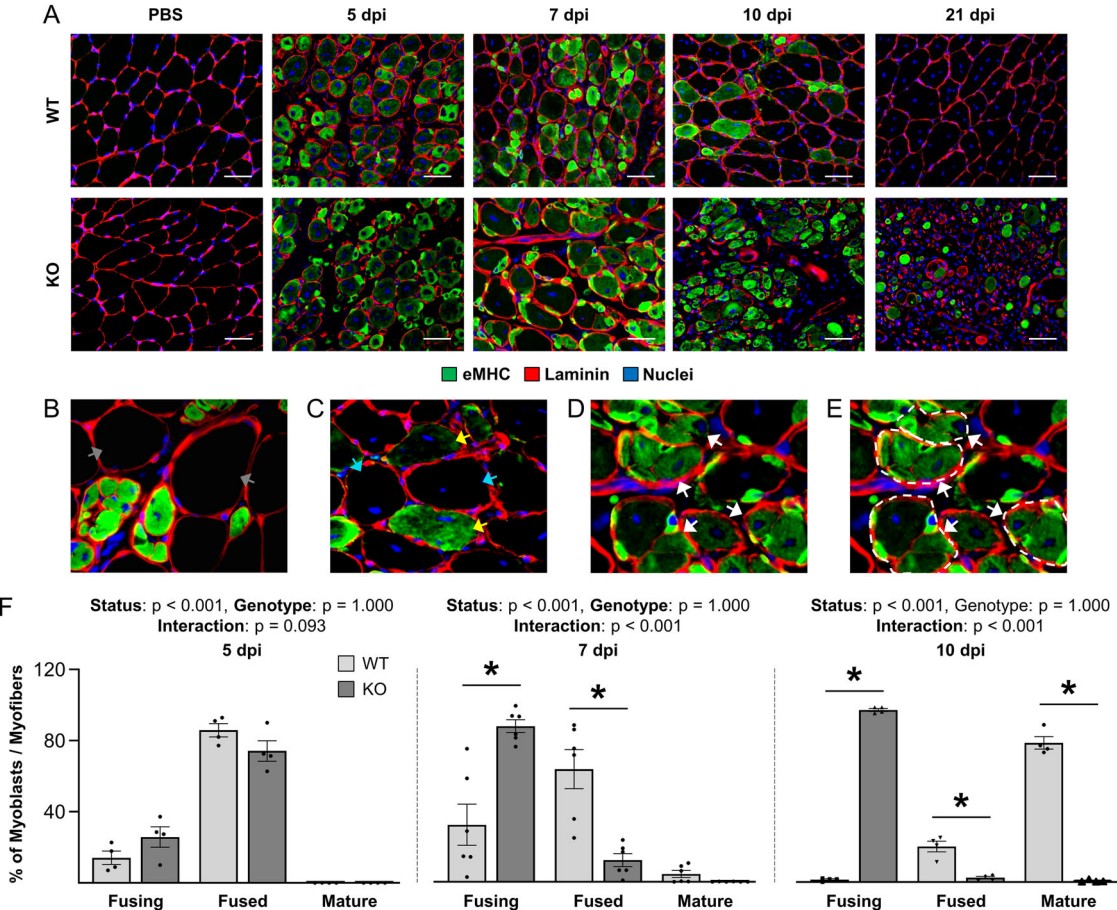

**Figure 4. The loss of TRIM28 in satellite cells leads to impaired fusion during injury-induced myogenesis/regeneration.**

Wild-type (WT) mice and tamoxifen-inducible satellite cell-specific TRIM28 knockout mice (KO) mice were treated with tamoxifen. At 14 days post tamoxifen, their tibialis anterior muscles were injected with $BaCl_2$ to induce injury or PBS as a control condition. (A) At 5, 7, 10, and 21 days post-injury (dpi), tibialis anterior muscles were collected and mid-belly cross-sections were subjected to immunohistochemistry for eMHC, laminin, and nuclei. (B–E) The myoblasts/myofibers in (A) were categorized as being in one of the following stages: fusing, fused, mature, or pre-existing. (B) Pre-existing (gray arrows) were defined as eMHC negative myofibers that did not contain centrally located nuclei and were likely not damaged by $BaCl_2$. (C) Mature (cyan arrows) were defined as eMHC-negative myofibers with centrally located nuclei. Fused (yellow arrows) were defined as eMHC-positive myoblasts/myofibers that did not contain interior laminin borders. (D, E) Fusing (white arrows) were defined as myoblasts/myofibers that contained clusters of eMHC positive cells that possessed individual laminin borders and were surrounded by a thicker outer laminin border (white dashed line in E). (F) Proportion of myoblasts/myofibers at 5, 7, and 10 dpi that were at the stage of fusing, fused, or mature (5 dpi: two-way ANOVA, $n = 4$/group; 7 dpi: two-way ANOVA, $n = 6$/group, *$p < 0.0001$; 10 dpi: two-way ANOVA, $n = 4$/group, *$p < 0.0001$). Values are means ± SEM, * indicates a significant difference from WT within the same category of myoblasts/myofibers, $p < 0.05$. Scale bars = 50 μm. Source data are available online for this figure.

Qualitatively, the impaired regenerative ability of the KO muscles became readily apparent by 10 dpi, and since the loss of TRIM28 had previously been reported to inhibit the differentiation-induced expression of MYOG (Singh et al, 2015), we reasoned that the impaired regenerative response might be due to a failure in the ability of the satellite cells to differentiate into myoblasts (i.e., become MYOG positive). However, as mentioned above, the presence of eMHC serves as a marker of differentiated myoblasts, and no apparent difference in the number of eMHC-positive myoblasts/myofibers was observed at 5 and 7 dpi in the WT versus KO muscles. Moreover, we found that the number of MYOG-positive nuclei per unit area at 10 dpi in KO muscles was actually 14-fold higher than in WT muscles (Appendix Fig. S3A,B). Accordingly, our results provided further support for the notion that TRIM28 is not required for the differentiation of satellite cells.

During regeneration, differentiated myoblasts align with each other and then fuse to form new myofibers. After the new myofibers have been formed, additional fusion events will occur until the new myofibers have grown to a mature size (Millay, 2022). As mentioned above, one of the first notable differences between the WT and KO muscles was that the size of the myoblasts/myofibers in the WT muscles increased between 7 and 10 dpi, but the increase in size did not occur in the KO muscles. Based on this observation, as well as our previous results with the SA model, we suspected that the loss of TRIM28 might led to a defect at the level of fusion. Hence, to further pursue this, we looked more closely at muscles 7 dpi and found that both WT and KO muscles were highly enriched with clusters of well-aligned eMHC-positive myoblasts/myofibers that were surrounded by a thick outer layer of laminin. Importantly, however, in the KO muscles, a much higher proportion of these clusters contained myoblasts/myofibers with a

dense wall of filamentous actin between the aligned cells (Appendix Fig. S4). Previous studies have shown that, prior to fusion, dense walls of filamentous actin appear between the aligned cells, and then, as fusion progresses, the walls of filamentous actin break down (Duan and Gallagher, 2009; Duan et al, 2018). Accordingly, the results in Appendix Fig. S4 suggested that the loss of TRIM28 may have inhibited the ability of the myoblasts to either engage in and/or complete the process of fusion.

To more thoroughly characterize the putative fusion defect, myoblasts/myofibers at 5, 7, and 10 dpi were categorized as being in one of the following stages: fusing, fused, mature, or pre-existing (Fig. 4B–E), and then the proportion in each stage was determined. As shown in Fig. 4F, all of the myoblasts/myofibers in WT and KO muscles at 5 dpi were categorized as "fusing" or "fused". By 10 dpi, the myoblasts/myofibers in WT muscles had transitioned to the point at which >75% were categorized as "mature" myofibers. However, in the KO muscles, almost no "mature" myofibers were identified, instead, >96% of the myoblasts/myofibers were still categorized as "fusing". Thus, these results provided further evidence for the notion that the loss of TRIM28 inhibits the ability of the myoblasts/myofibers to complete the process of fusion that occurs between 7 and 10 dpi. Of note, previous studies have reported that myoblasts will die if they do not undergo fusion (Bi et al, 2018; Millay et al, 2014), and given the results in Fig. 4F, this likely explains why KO muscles at 21 dpi were largely devoid of myoblasts/myofibers. To further investigate the fate of these myoblasts/myofibers, WT and KO muscle cross-sections were subjected to a TUNEL assay to determine whether there was indeed a significant difference in the number of TUNEL-positive nuclei between the genotypes. The TUNEL assay indicates fragmented DNA which is a marker of cell death and, as shown in Appendix Fig. S5, there were significantly more TUNEL-positive nuclei in the KO muscles than the WT at 10 dpi. Further, some of these TUNEL-positive nuclei were inside of Laminin+ and multinucleated nascent myofibers (Appendix Fig. S5A). This provides evidence to support the notion that the myoblasts/myofibers that did not undergo fusion died thereby contributing to the observed phenotype and overall lack of regeneration.

## The loss of TRIM28 in satellite cells leads to a profound functional deficit and excessive fibrosis following BaCl₂-induced injury

The above results indicated that, at the cellular level, injury-induced myogenesis/regeneration is severely impaired by the satellite cell-specific loss of TRIM28. To determine whether this impairment translated into a functional deficit, the mass and force-producing capabilities of the muscles were measured. As shown in Fig. 5A–D, the outcomes revealed that, in WT muscles, peak tetanic force production was 64% lower at 7 dpi and this loss was restored to control levels by 21 dpi. However, in KO muscles, force production was 95% lower at 7 dpi, and very little, if any, recovery was observed at 21 dpi. In line with these results, we also found that the mass and gross appearance of the WT muscles at 21 dpi were very similar to the control muscles, but the KO muscles were markedly small and pale in color (Fig. 5E).

The pale color of the KO muscles suggested that they might suffer from excessive fibrosis (Sihvo et al, 2014), and this was substantiated by Sirius Red staining which revealed that the

proportion of the muscles that were occupied by collagen progressively increased following injury in the KO mice (Fig. EV2A,B). This was intriguing because excessive fibrosis has been widely associated with aberrant regeneration (Mann et al, 2011). Moreover, previous studies have reported that the secretion of miR-206 by satellite cells can inhibit the expression of collagen by fibroblasts and that TRIM28 is required for the differentiation-induced expression of miR-206 (Fry et al, 2017; Singh et al, 2015). Thus, we envisioned that the satellite cell-specific loss of TRIM28 might predispose the KO muscles to a miR-206-dependent increase in fibrosis and, in turn, inhibit the ability of the myoblasts/myofibers to progress through the process of fusion. Accordingly, we tested this by treating the mice with Nilotinib, which is a drug that has been shown to reduce fibrosis in chronically injured muscles (Lemos et al, 2015). As reported in Fig. EV2C,D, Nilotinib led to a 50% reduction in the area of the KO muscles that were occupied by collagen at 10 dpi; however, the proportion of myoblasts/myofibers that had progressed beyond the stage of "fusing" was only slightly improved (Fig. EV2E,F). Thus, although our data indicates that the satellite cell-specific loss of TRIM28 leads to excessive fibrosis, this effect does not appear to be a major driver of the impaired fusion that is observed in KO muscles.

## The loss of TRIM28 in primary myoblasts leads to a fusion defect

To more clearly define the role that TRIM28 plays in fusion, we isolated primary myoblasts from WT and KO mice. After confirming that the primary myoblasts from the KO mice were negative for TRIM28 (Fig. EV3A,B), they were subjected to a myotube formation assay and the differentiation index (i.e., the proportion of nuclei that were inside myosin heavy chain (MHC) positive cells) and fusion index (i.e., the proportion of nuclei that were inside MHC positive multinucleated cells) were measured. As shown in Fig. EV3C–E, the outcomes revealed that the KO primary myoblasts had a slightly lower differentiation index along with a dramatically lower fusion index. These observations were consistent with our in vivo results and further suggested that TRIM28 plays an important role in the process of fusion. However, for reasons that are not clear, the KO primary myoblasts were difficult to propagate and this prevented us from performing additional mechanistic studies (Fig. EV3F).

To overcome the propagation defect we switched to a shRNA-mediated knockdown system. Specifically, WT primary myoblasts were infected with lentivirus encoding a scrambled shRNA (CNT) or shRNA targeting *Trim28* mRNA (KD). As shown in Fig. 6A,B, infection with KD shRNA led to a clear reduction in the protein level of TRIM28 when the cells were cultured in growth media and this effect was maintained after the cells had been induced to differentiate for 2 days. In agreement with our in vivo analyses, we also found that the knockdown of TRIM28 did not significantly alter the expression of MYOD (Fig. 6C). However, the knockdown of TRIM28 did lead to a trend towards a decrease in differentiation-induced expression of MYOG (Fig. 6D). Moreover, our myotube formation assay revealed that the knockdown of TRIM28 did not significantly impact the differentiation index but, once again, it did lead to a severe reduction in the fusion index (Fig. 6E–G). Finally, as shown in Fig. 6H, we also determined that the cells infected with KD shRNA did not suffer from a propagation defect, and therefore,

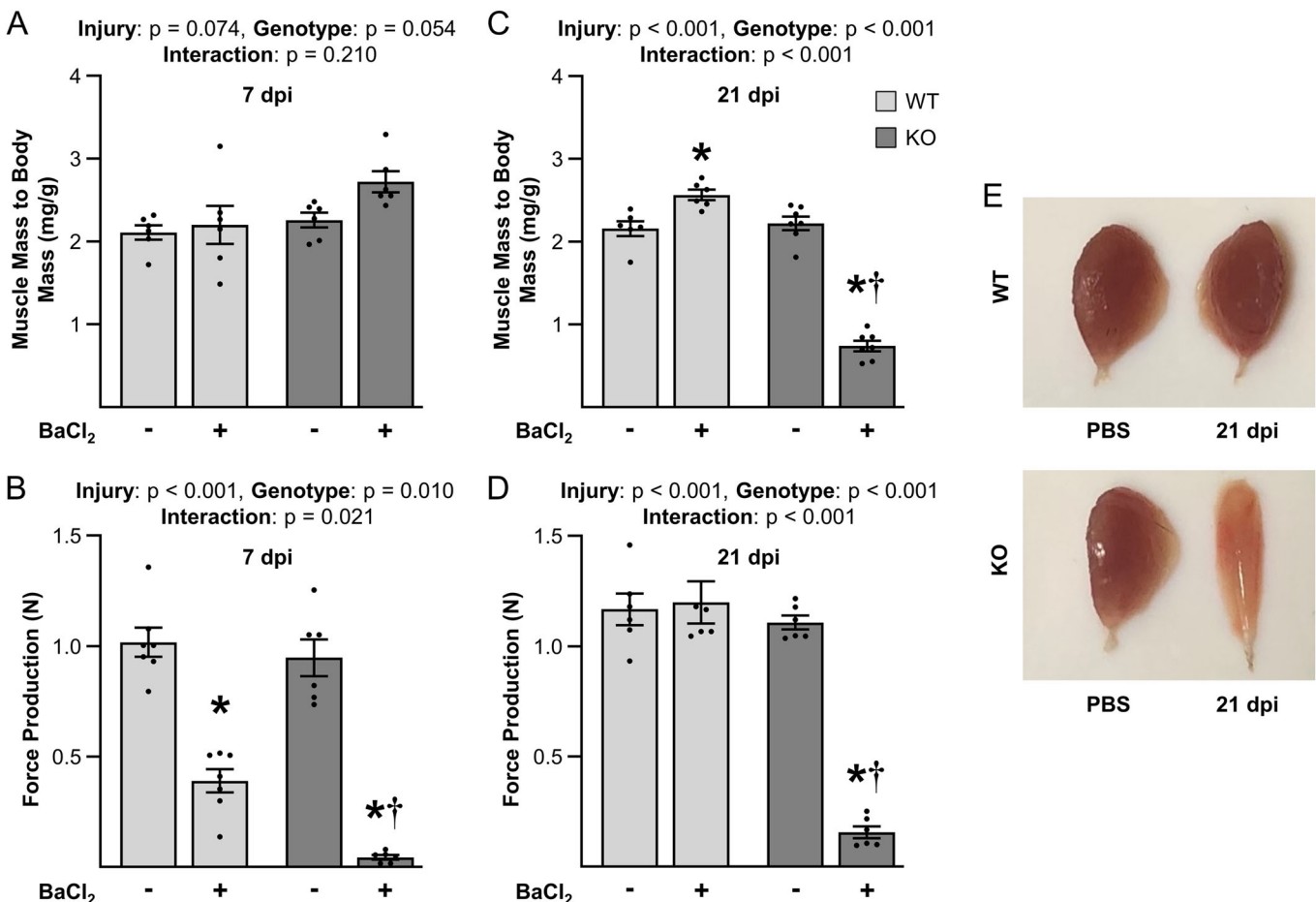

**Figure 5. The loss of TRIM28 in satellite cells leads to a profound functional deficit following BaCl₂-induced injury.**

Wild-type (WT) mice and tamoxifen-inducible satellite cell-specific TRIM28 knockout mice (KO) mice were treated with tamoxifen. At 14 days post tamoxifen, their tibialis anterior muscles were injected with BaCl₂ (+) to induce injury or PBS (−) as a control condition. At 7 days post-injury (dpi), (A) the muscle mass to body mass ratio (two-way repeated-measures ANOVA, $n = 6$/group) and (B) in situ peak tetanic force production were measured (two-way repeated-measures ANOVA, $n = 6$–7/group, $*p < 0.001$, $†p < 0.001$). At 21 dpi, (C) the muscle mass to body mass ratio (two-way repeated-measures ANOVA, $n = 6$–7/group, $*p = 0.0002$ or $<0.0001$, $†p < 0.0001$) and (D) in situ peak tetanic force production were measured (two-way repeated-measures ANOVA, $n = 6$/group, $*$ and $†p < 0.0001$). (E) The gross appearance of tibialis anterior muscles at 21 dpi. Values are group means ± SEM. $*$ indicates a significant effect of injury, $†$ indicates a significant difference between the BaCl₂ treated groups, $p < 0.05$. Source data are available online for this figure.

we were confident that the knockdown system would enable us to gain further insight into why TRIM28 is necessary for myoblasts to either engage in and/or complete the process of fusion.

## TRIM28(S473) phosphorylation is not required for myoblast fusion in vitro

Collectively, our in vivo and in vitro analyses strongly suggested that TRIM28 is not required for myoblast differentiation (Figs. 2C,D, 4A, 6E,F). Yet, TRIM28 clearly played a major role in the regulation of myogenesis, and it appeared that this was due to its ability to regulate fusion. Consistent with this point, it was previously reported that the loss of TRIM28 significantly inhibits myotube formation in C2C12 myoblasts: however, the same study also concluded that TRIM28 exerts its functions via the regulation of differentiation (Singh et al, 2015). For instance, using the same shRNA constructs that we employed, it was determined that the

knockdown of TRIM28 in C2C12 myoblasts inhibited the differentiation-induced increase in the expression of critical regulators of differentiation such as MYOG. Moreover, it was concluded that the ability of TRIM28 to regulate differentiation was dependent on TRIM28(S473) phosphorylation. This is important because, as shown in Fig. 1, our interest in TRIM28 began with the observation that myogenic stimuli, such as mechanical loading, lead to a very robust increase in TRIM28(S473) phosphorylation. As such, we wanted to determine whether the role that TRIM28 plays in the regulation of myogenesis/fusion is dependent on changes in S473 phosphorylation. Thus, to address this, WT primary myoblasts were subjected to a rescue-based experiment in which shRNA-mediated knockdown of endogenous TRIM28 was coupled with "rescue" expression of an shRNA-resistant form of wild-type TRIM28 (KD + WT) or an shRNA-resistant and phosphodefective form of TRIM28 (KD + PD) in which the serine 473 residue had been mutated to a non-phosphorylatable alanine.

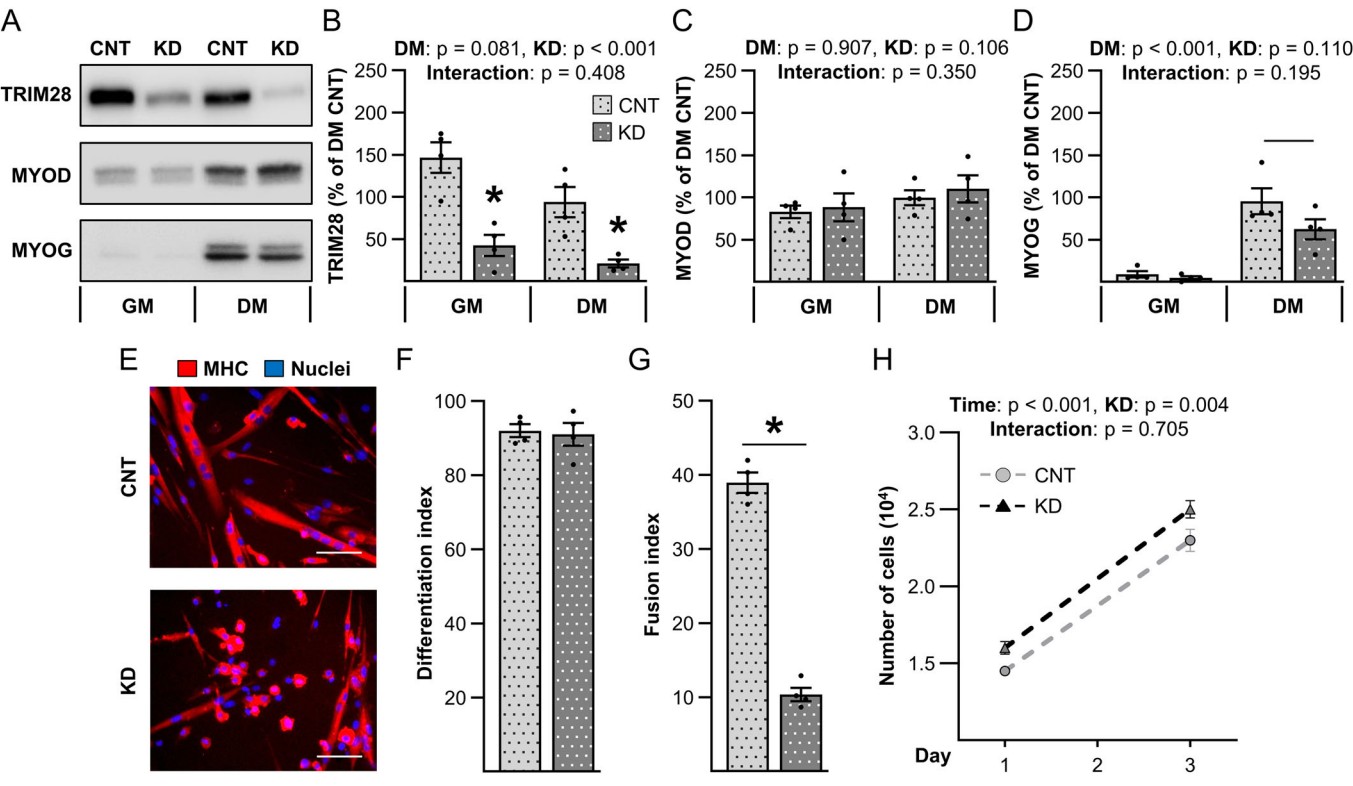

**Figure 6.  Knockdown of TRIM28 leads to a fusion defect during in vitro myotube formation.**

(A–D) Primary myoblasts were infected with lentivirus expressing scrambled shRNA (CNT) or shRNA targeting Trim28 mRNA (KD). The infected cells were cultured in growth media (GM) or differentiation media (DM) for 2 days and subjected to western blot analysis of TRIM28 (two-way repeated-measures ANOVA, $n = 4$/group, *$p < 0.001$ or $= 0.005$), MYOD (two-way repeated-measures ANOVA, $n = 4$/group), and MYOG (two-way repeated-measures ANOVA, $n = 4$/group). (E–G) Infected myoblasts were subjected to a myotube formation assay and immunohistochemistry for MHC and nuclei. (F) The differentiation index (% of nuclei inside MHC positive cells) (unpaired Student's t-tests, $n = 4$/group), and (G) the fusion index (% of nuclei inside MHC positive multinucleated cells) were quantified (unpaired Student's t-tests, $n = 4$/group, *$p < 0.0001$). (H) $2 \times 10^4$ infected myoblasts were seeded on day 0 and quantified on days 1 and 3 (two-way repeated-measures ANOVA, $n = 4$/group). Values are group means ± SEM, each sample representing an independent line of isolated primary myoblasts. * indicates a significant effect of genotype within a given condition. $P < 0.05$. Horizontal line indicates a significant effect of DM. Scale bars = 50 μm. Source data are available online for this figure.

The rescue-based approach that we employed utilized the same lentiviral constructs that were used in a previous study with C2C12 myoblasts (Singh et al, 2015), and the first thing we noticed was that the transduction efficiency of the "rescue" constructs (i.e, KD + WT and KD + PD) was quite low when compared with the efficiency of the CNT and KD constructs. Nevertheless, we were able to culture enough myoblasts to conduct myotube formation assays and, as expected, expression of the shRNA-resistant form of wild-type TRIM28 (i.e., KD + WT) was able to rescue the fusion defect that was observed in the KD condition (Appendix Fig. S6). However, much to our surprise, expression of the shRNA-resistant and phosphodefective form of TRIM28 (i.e., KD + PD) also rescued the fusion defect (Appendix Fig. S6).

The results in Appendix Fig. S6 suggested that TRIM28(S473) phosphorylation is not required for the role that TRIM28 plays in the regulation of fusion and this is very different from the observations that were reported in C2C12 myoblasts (Singh et al, 2015). We were puzzled by this discrepancy, and since both of our studies employed the same constructs, we reasoned that the discrepancy might be due to the fact that we employed primary myoblasts. Hence, to test this, we tried to repeat our studies in C2C12 myoblasts. However, just like with the primary myoblasts,

we found that the transduction efficiency of the KD + WT and KD + PD constructs was very low. Indeed, despite several attempts, the low transduction efficiency prevented us from being able to culture enough C2C12 myoblasts to conduct myotube formation assays. As such, we were not able to establish whether the discrepancy in our observations could be attributed to differences in C2C12 myoblasts versus primary myoblasts.

## A novel in vivo "rescue" approach reveals that TRIM28(S473) phosphorylation in satellite cells is not required for injury-induced myogenesis/regeneration

Given the difficulties that were encountered with the in vitro rescue-based experiments, we were only moderately convinced that TRIM28(S473) phosphorylation does not play a role in the regulation of fusion. Moreover, since the shRNA-mediated knockdown studies relied on the use of in vitro conditions, they could not provide definitive insights into the role that TRIM28(S473) phosphorylation plays in vivo. Hence, to overcome these limitations, we developed an innovative genome editing strategy that enabled us to perform rescue-based experiments in vivo. Specifically, we created mice that allowed for tamoxifen-inducible and satellite cell-specific knockout of

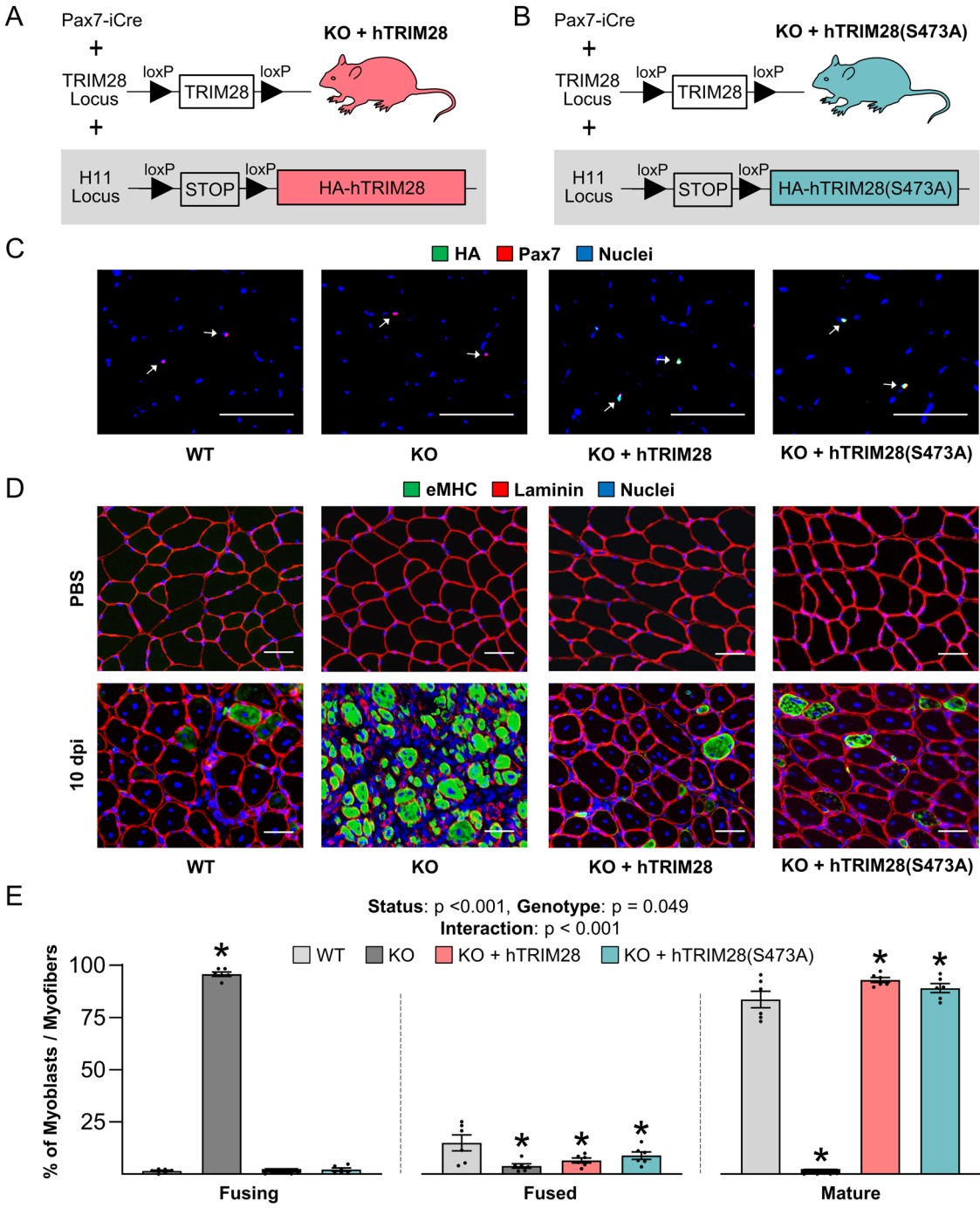

**Figure 7. TRIM28(S473) phosphorylation in satellite cells is not required for injury-induced fusion in vivo.**

Strategy for creating mice that allowed for tamoxifen-inducible and satellite cell-specific knockout of endogenous TRIM28 along with tamoxifen-inducible and satellite cell-specific "rescue" expression of (A) human TRIM28 (KO + hTRIM28) or (B) a S473A phosphodefective mutant of hTRIM28 (KO + hTRIM28(S473A)). (C) Wild-type (WT), tamoxifen-inducible satellite cell-specific TRIM28 knockout (KO), KO + hTRIM28, and KO + hTRIM28(S473A) mice were treated with tamoxifen. At 14 days post-tamoxifen, mid-belly cross sections from the tibialis anterior muscles were subjected to immunohistochemistry for Pax7, HA, and nuclei. White arrows indicate satellite cells. (D) At 14 days post tamoxifen tibialis anterior muscles were injected with $BaCl_2$ to induce injury or PBS as a control condition. At 10 days post-injury (dpi), mid-belly cross sections of the tibialis anterior muscles were subjected to immunohistochemistry for eMHC, laminin, and nuclei. As defined in Fig. 4, myoblasts/myofibers in the injured muscles were assigned to one of the following four stages: fusing, fused, mature, or pre-existing. (E) Proportion of myoblasts/myofibers that were at the stage of fusing, fused, or mature (two-way repeated-measures ANOVA, $n = 6$/group, *$p < 0.001$, $= 0.001$, $= 0.004$, $= 0.009$, $= 0.029$, or $= 0.049$). Values are presented as the group mean ± SEM. Data were analyzed with two-way repeated-measures ANOVA. * indicates a significant difference from WT, $p < 0.05$. Scale bars $= 50\ \mu m$. Source data are available online for this figure.

endogenous TRIM28 along with simultaneous satellite cell-specific and tamoxifen-inducible "rescue" expression of human TRIM28 (KO + hTRIM28) or a S473A phosphodefective mutant of hTRIM28 (KO + hTRIM28(S473A)) (Figs. EV4 and 7A,B).

To create the KO + hTRIM28 mice, a construct encoding HA-tagged hTRIM28 (HA-hTRIM28) downstream of a LoxP-Stop-LoxP (LSL) cassette was knocked into the H11 locus of TARGATT™ C57BL6 mice (Chen-Tsai, 2019) (Fig. EV4A,B). These mice were then crossed with KO mice to create KO + hTRIM28 progeny with a genotype of *Pax7*-iCre$^{+/-}$ : *Trim28*$^{flox/flox}$ : LSL-HA-hTRIM28$^{+/+}$. To create the KO + hTRIM28(S473A) mice, CRISPR-Cas9-mediated homologous-directed repair was used to make a single point mutation in hTRIM28 of the TARGATT LSL-HA-hTRIM28 mice. The single point mutation converted the serine 473 residue of hTRIM28 to a non-phosphorylatable alanine, and these mice were crossed with KO mice to create KO + hTRIM28(S473A) progeny with a genotype of *Pax7*-iCre$^{+/-}$: *Trim28*$^{flox/flox}$ : LSL-HA-hTRIM28(S473A)$^{+/+}$ (Fig. EV4C,D). The single point mutation of F0 and F1 founders was confirmed through amplicon sequencing followed with CRISPResso2 analysis.

To assess whether the "rescue" expression of HA-hTRIM28 in the KO + hTRIM28 and KO + hTRIM28(S473A) mice was working appropriately, TA muscles were collected 14 days after mice had been treated with tamoxifen and assessed for the presence of HA-positive satellite cells. As expected, HA-positive satellite cells were widely abundant in the muscles from KO + hTRIM28 and KO + hTRIM28(S473A) mice, but not in the muscles from WT and KO mice (Fig. 7C). With these results in hand, we next set out to establish whether the expression of hTRIM28 and/or hTRIM28(S473A) could rescue the fusion defect that occurs in muscles of KO mice at 10 days after BaCl$_2$-induced injury. Consistent with the results presented in Fig. 4, we found the vast majority of the myoblasts/myofibers in WT muscles were categorized as "mature" myofibers at 10 dpi, whereas almost no "mature" myofibers were identified in the muscles of the KO mice (Fig. 7D,E). Importantly, the expression of hTRIM28 in the KO mice completely rescued this defect, and the same point was true for the expression of hTRIM28(S473A).

The results in Fig. 7 reinforced the validity of the observations that were made with our in vitro rescue-based studies, and further demonstrated that changes in TRIM28(S473) phosphorylation are not required for the role that TRIM28 plays in the regulation of fusion. To extend these observations we also assessed whether any of the roles that TRIM28 plays in myogenesis/regeneration are dependent on S473 phosphorylation (e.g., restoration of myofiber CSA, force production, etc.). Specifically, TA muscles from WT, KO, KO + hTRIM28, and KO + hTRIM28(S473A) mice were injected with BaCl$_2$ or PBS and collected at 21 dpi. Consistent with the results in Fig. 5, we found that injured KO muscles were small and pale in color at 21 dpi, but such effects were not observed in the injured muscles of the KO + hTRIM28 or KO + hTRIM28(S473A) mice (Fig. 8A,B, see Fig. 5E for representative images of WT and KO muscles). Cross-sections of the TA muscles also revealed that, aside from the presence of HA-positive myonuclei, the regenerated myofibers in KO + hTRIM28 and KO + hTRIM28(S473A) muscles were indistinguishable from those of WT muscles (Figs. 8C and EV5). Moreover, the force-producing ability of the muscles from KO mice was severely impaired at 21 dpi, but again, this effect was not observed in the muscles from KO + hTRIM28 or KO + hTRIM28(S473A) mice (Fig. 8D). Thus, our novel in vivo "rescue" approach not only enabled us to establish that TRIM28(S473)

phosphorylation in satellite cells is not required for any of the events that are necessary for successful myogenesis/regeneration of skeletal muscle, but it also provided further support for the validity of our in vitro observations.

## The loss of TRIM28 inhibits the process of fusion pore formation but not hemifusion

Having established that the primary conclusions from the in vitro assays were conserved in vivo, we next sought to gain additional insight into the mechanism via which TRIM28 regulates fusion. To accomplish this, we turned to one of the more widely accepted models in the field which argues that fusion occurs in two major stages. Specifically, the first stage is referred to as hemifusion and involves the mixing of the outer membranes of the fusing cells (Chernomordik and Kozlov, 2005; Leikina et al, 2018; Millay, 2022), whereas the second stage involves the formation of a fusion pore that allows for the mixing of the intracellular contents of the fusing cells (Golani et al, 2021; Leikina et al, 2018). Thus, to determine whether the loss of TRIM28 affected hemifusion and/or fusion pore formation, CNT and KD myoblasts were differentiated for 24 h, detached, and then briefly labeled with either an intracellular content probe or a lipid membrane probe (Fig. 9A). The separately labeled cells were then mixed, plated, and allowed to differentiate. After 30 h, the cells were analyzed and mononucleated cells that were positive for both probes were classified as having only undergone hemifusion, whereas all multinucleated cells were classified as having undergone fusion pore formation. As illustrated in Fig. 9A–D, the results of our analyses revealed that the loss of TRIM28 did not significantly alter the proportion of the nuclei that resided within cells that had only undergone hemifusion, but it did lead to a highly significant reduction in the proportion of nuclei that resided within cells that had undergone fusion pore formation.

Previous studies have identified two muscle-specific fusogenic proteins that play a critical role in fusion and these proteins are generally referred to as myomaker (MYMK, also known as TMEM8C) and myomixer (MYMX, also known as myomerger and MINION) (Bi et al, 2017; Millay et al, 2013; Quinn et al, 2017; Zhang et al, 2017). Recent in vitro studies have shown that MYMK is required for hemifusion and that fusion pore formation will not occur if either of the two fusing cells lacks it (Leikina et al, 2018; Millay, 2022; Zhang et al, 2020). On the other hand, MYMX is thought to physically drive the process of fusion pore formation, and unlike with MYMK, fusion pore formation is only partially inhibited if either of the two fusing cells lacks MYMX (Zhang et al, 2020; Zhang et al, 2017). Since the results in Fig. 9A–D indicated that the loss of TRIM28 impairs fusion pore formation but not hemifusion, we suspected that the loss of TRIM28 would regulate fusion pore formation in a manner that is analogous with the functions of MYMX (i.e., fusion pore formation would only be partially inhibited if one of the two fusing cells lacked it). Hence, to test this, CNT and KD primary myoblasts were differentiated for 24 h, detached, and then briefly labeled with a red- or green-fluorescent intracellular content probe. The separately labeled cells were then mixed into one of the following types of pairs: CNT-red with CNT-green, CNT-red with KD-green, or KD-red with KD-green (Fig. 9E,F). Fusion pore formation as a result of heterologous fusion was then quantified by identifying multinucleated cells that were positive for both of the intracellular content probes. As shown

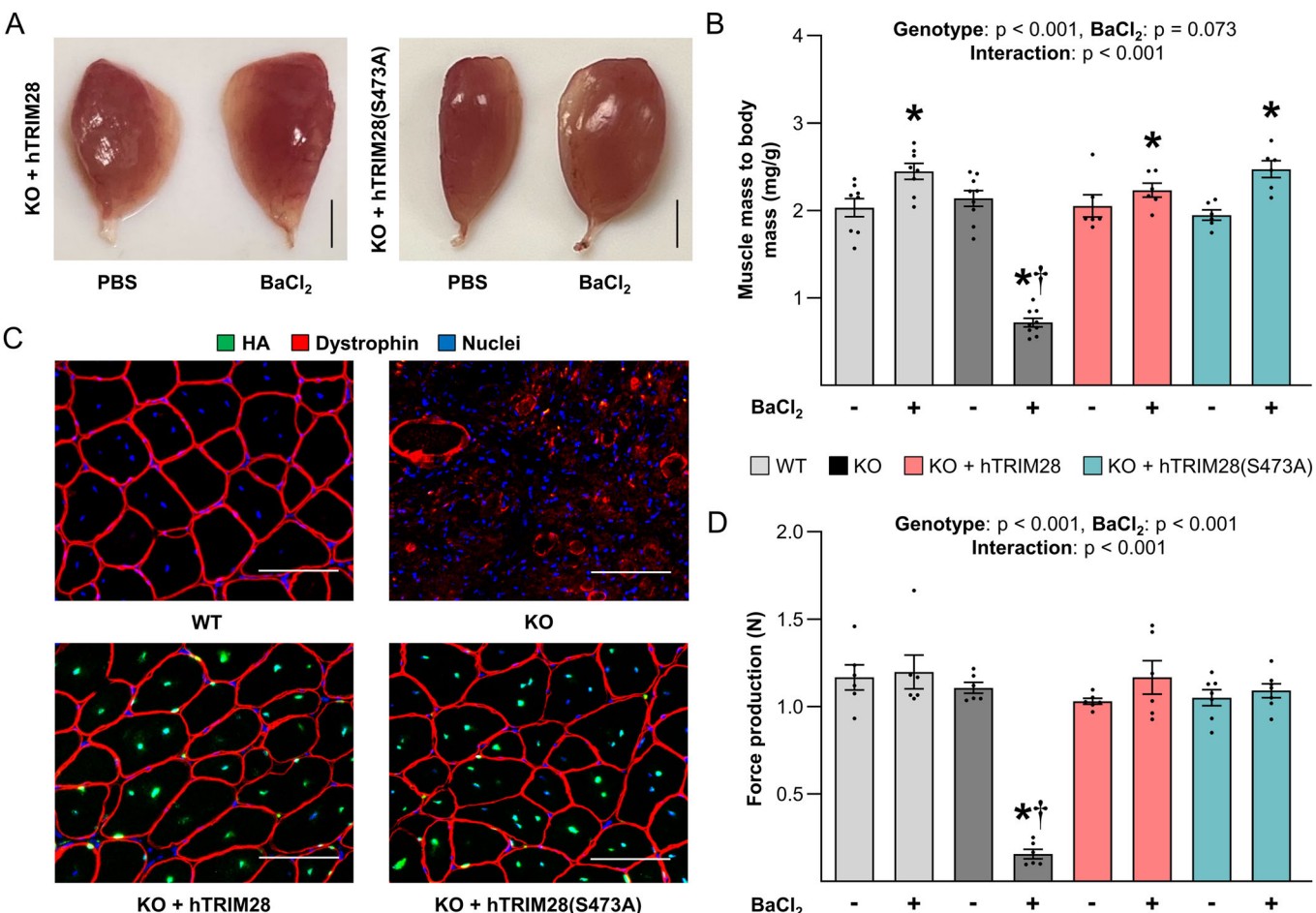

**Figure 8. TRIM28(S473) phosphorylation in satellite cells is not required for injury-induced myogenesis/regeneration.**

At 14 days post tamoxifen the tibialis anterior muscles of wild-type (WT) mice, tamoxifen-inducible and satellite cell-specific TRIM28 knockout mice (KO) mice, as well as KO mice that contain tamoxifen-inducible "rescue" expression of hTRIM28 (KO + hTRIM28) or phosphodefective hTRIM28 (KO + hTRIM28(S473A)) were injected with $BaCl_2$ (+) to induce injury or PBS (−) as a control condition. The tibialis anterior muscles were collected at 21 days post-injury. (A) Gross appearance of tibialis anterior muscles. Scale bars = 2 mm (see Fig. 5 for representative images of the WT and KO muscles). (B) The muscle mass to body mass ratios (two-way repeated-measures ANOVA, $n = 6$–9/group, $*p < 0.0001$ or $= 0.046$, $\dagger p < 0.0001$). (C) Mid-belly cross sections of the muscles were subjected to immunohistochemistry for the HA tag, dystrophin, and nuclei. Scale bars = 50 μm. (D) In situ peak tetanic force production (two-way repeated-measures ANOVA, $n = 6$–7/group, * and † $p < 0.0001$). Values are presented as the group mean ± SEM. * indicates a significant effect of $BaCl_2$ within a given genotype, † indicates significant difference from the $BaCl_2$ treated WT condition, $p < 0.05$. Source data are available online for this figure.

in Fig. 9E–G, the magnitude of fusion pore formation was maximal when both cells expressed a normal level of TRIM28, it was partially inhibited when only one cell expressed a normal level of TRIM28, and it was largely abolished when both cells suffered from the loss of TRIM28. These outcomes are extremely similar to what has previously been reported with the loss of MYMX in human primary myoblasts and therefore suggested that TRIM28 regulates fusion via a MYMX-dependent mechanism (Zhang et al, 2020).

## The loss of TRIM28 inhibits the induction of MYMX expression during differentiation and injury-induced myogenesis

Previous studies have shown that the induction of differentiation leads to an increase in the expression of MYMX and MYMK, and the results in Fig. 9 revealed that the loss of TRIM28 inhibits the

process of fusion pore formation but not hemifusion (Bi et al, 2017; Millay et al, 2013; Quinn et al, 2017; Zhang et al, 2017). Accordingly, we suspected that the loss of TRIM28 would inhibit the differentiation-induced increase in MYMX expression, but not MYMK. To test this, lysates from the CNT and KD primary myoblasts described in Fig. 6A–D were subjected to western blot analysis for MYMX and MYMK. As shown in Fig. 10A–C, the results indicated that the knockdown-mediated loss of TRIM28 did not affect the differentiation-induced increase in MYMK. However, the differentiation-induced increase in MYMX was substantially reduced. We also discovered that the differentiation-induced increase in MYMX was very highly correlated with the level of TRIM28 in the cells (Fig. 10C,E), but no significant correlation was detected between the levels of TRIM28 and MYMK in the cells (Fig. 10B,D). To determine whether similar observations would be made in vivo, we examined the level of MYOD, MYOG, MYMK,

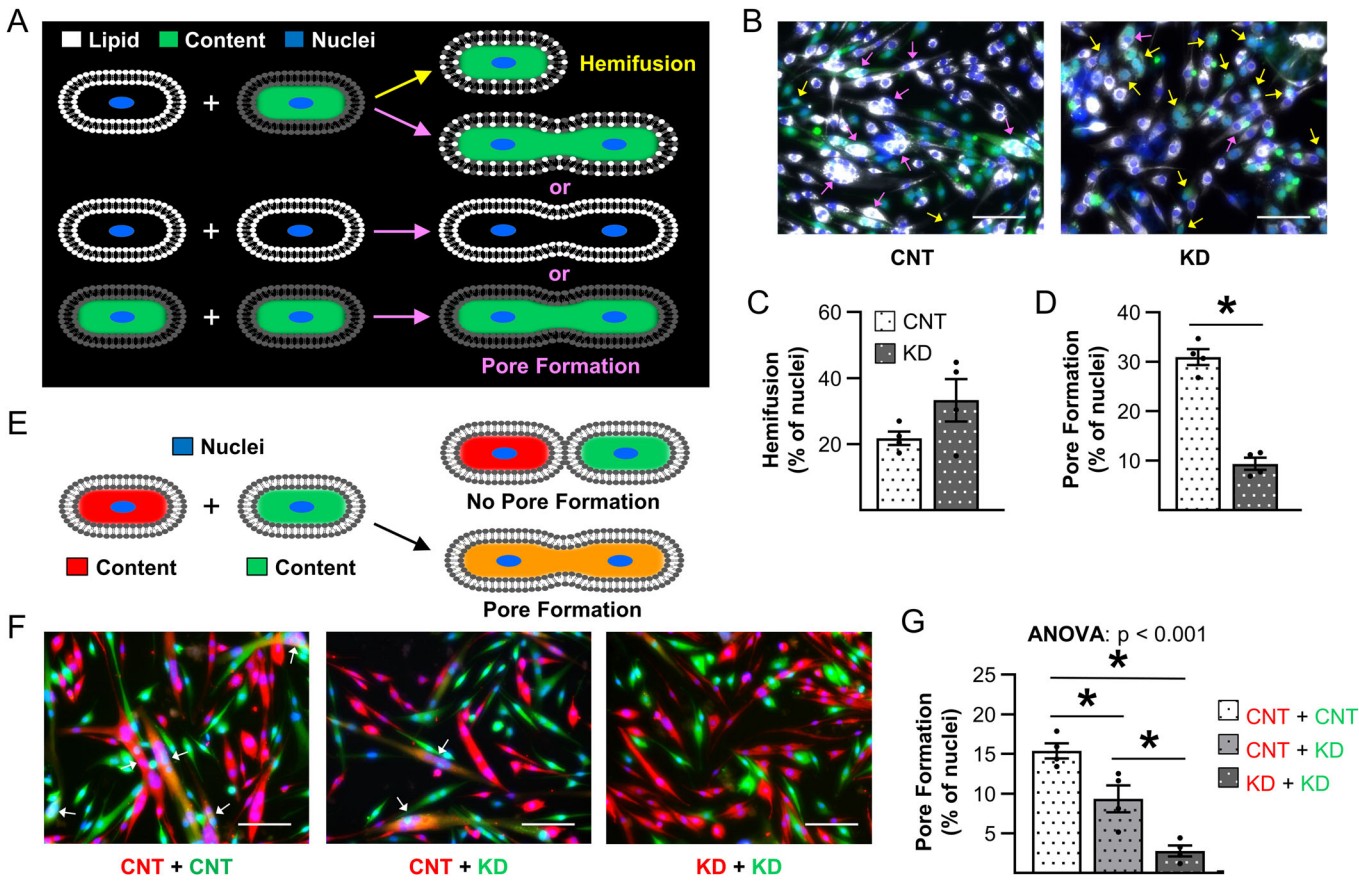

**Figure 9. The loss of TRIM28 inhibits the process of fusion pore formation but not hemifusion.**

Primary myoblasts were infected with a lentivirus expressing either scrambled shRNA (CNT) or shRNA targeting Trim28 (KD) and then differentiated for 24 h. (A, B) CNT and KD myoblasts were detached and briefly labeled with either an intracellular content probe or a lipid membrane probe. The separately labeled cells were then mixed and differentiated for an additional 30 h. Mononucleated cells that were positive for both probes were classified as having only undergone hemifusion (yellow arrows), whereas multinucleated were classified as having undergone fusion pore formation (pink arrows). (C, D) Quantification of the proportion of nuclei in cells that had undergone (C) only hemifusion, or (D) fusion pore formation (unpaired Student's t-test, n = 4/group, *p < 0.0001). (E, F) CNT and KD myoblasts were detached, separated into two sub-groups, and labeled with a red- or green-fluorescent intracellular content probe. The separately labeled cells were mixed as indicated in (F) and differentiated for an additional 24 h. Multinucleated cells that were positive for both probes (white arrows) were defined as having undergone fusion pore formation. (G) Quantification of the proportion of nuclei in cells that had undergone fusion pore formation during heterologous fusion (one-way ANOVA, n = 4/group, *p = 0.0144, = 0.0090, or <0.0001). Values are group means ± SEM, each sample representing an independent line of isolated primary myoblasts. * indicates a significant difference between the indicated groups, p < 0.05. Scale bars = 50 μm. Source data are available online for this figure.

and MYMX in the TA muscles of WT and KO mice that had been injured with $BaCl_2$. As shown in Fig. 10F–J, the outcomes were very similar to what was observed in the primary myoblasts with injury leading to an increase in all of the examined proteins. Specifically, the loss of TRIM28 did not affect the injury-induced increases in MYOD and MYMK, whereas the increase in MYOG was partially attenuated and the increase in MYMX was largely abolished. Thus, the results of our analyses, along with the a priori knowledge about MYMX, have led us to the conclusion that TRIM28 regulates myogenesis by controlling the induction of MYMX expression and concomitant fusion pore formation.

## Discussion

The overarching goal of this study was to determine whether TRIM28 is a critical regulator of myogenesis. This was viewed as an important

goal because myogenesis plays a vital role in the regulation of skeletal muscle mass, and the maintenance of muscle mass is essential for mobility, breathing, whole-body metabolism, and overall quality of life (Izumiya et al, 2008; Seguin and Nelson, 2003). Moreover, skeletal muscles are one of the few mammalian tissues that are capable of regenerating, and the process of regeneration recapitulates many aspects of embryonic myogenesis (Iismaa et al, 2018; Musarò, 2014). Indeed, impairments in myogenesis/regeneration are associated with the progression of diseases such as muscular dystrophy and contribute to the age-related decline in the ability of skeletal muscle to recover from injury (Munoz-Canoves et al, 2020; Yanay et al, 2020). Thus, advancements in our understanding of the mechanisms that regulate myogenesis might not only lead to the development of therapies for treating these conditions, but they could also provide pivotal insights for the broader field of regenerative medicine.

By using a variety of in vitro and in vivo models we established that, in satellite cells, TRIM28 plays a critical role in the regulation of

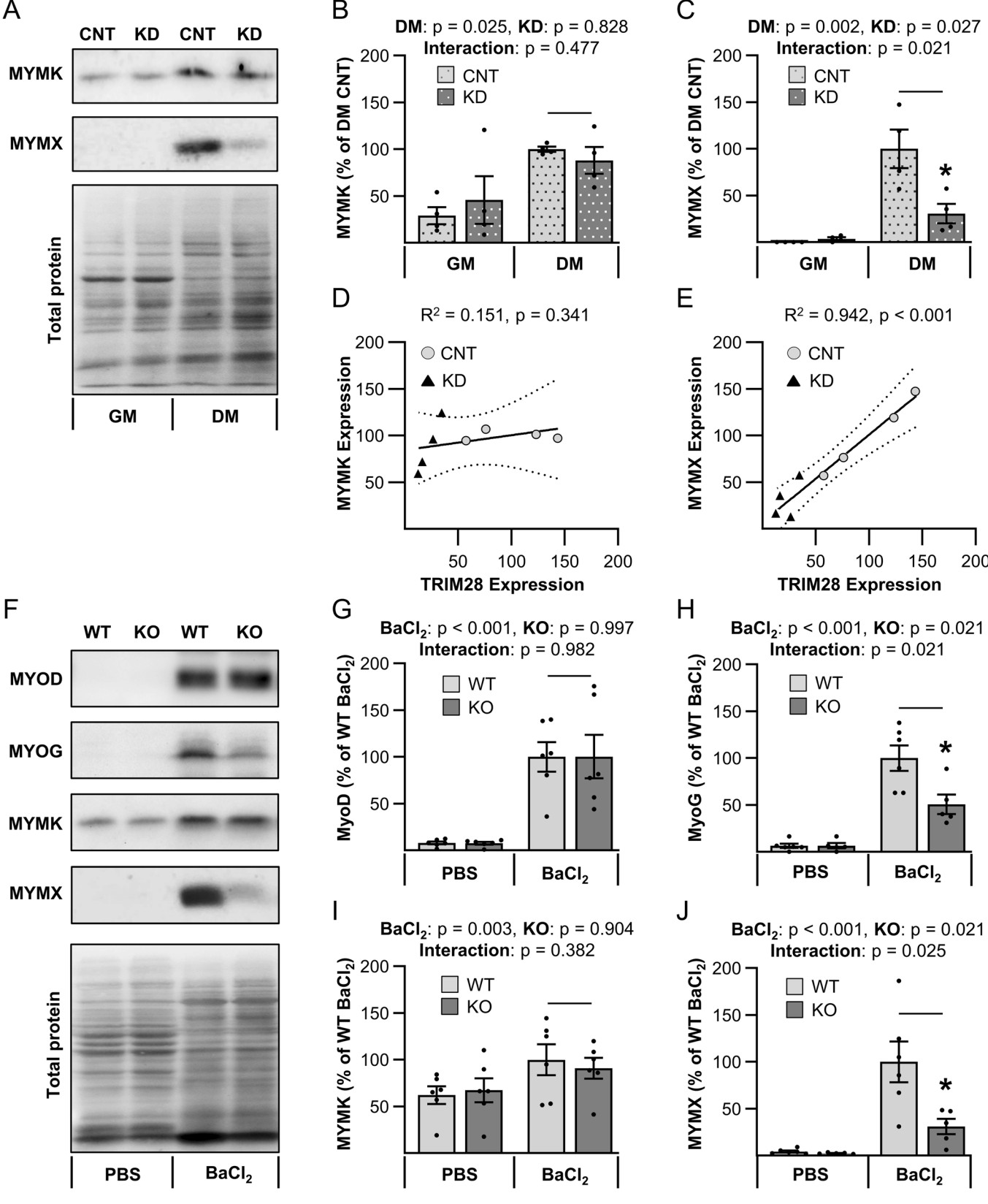

◄  **Figure 10.    The loss of TRIM28 inhibits the induction of MYMX expression during differentiation and injury-induced myogenesis.**

(A–C) As in Fig. 6, primary myoblasts were infected with lentivirus expressing scrambled shRNA (CNT) or shRNA targeting Trim28 mRNA (KD). The infected cells were cultured in growth media (GM) or differentiation media (DM) for 2 days and subjected to western blot analysis of myomaker (MYMK) (two-way repeated-measures ANOVA, $n = 4$/group) and myomixer (MYMX) (two-way repeated-measures ANOVA, $n = 4$/group, *$p < 0.0011$). (D, E) Simple linear regression analysis of TRIM28 and MYMK expression (D) or MYMX (E) from the infected cells that had been cultured in DM for 2 days (dashed lines represent the 95% confidence intervals). See Fig. 6 for the western blot analysis of TRIM28. (F–J) Tibialis anterior muscles from wild-type (WT) and tamoxifen-inducible satellite cell-specific TRIM28 knockout (KO) mice were injected with BaCl$_2$ to induce injury or PBS as a control condition. At 3 days post-injury the muscles were subjected to western blot analysis of MYOD (two-way repeated-measures ANOVA, $n = 6$/group), MYOG (two-way repeated-measures ANOVA, $n = 5$–6/group, *$p = 0.001$), MYMK (two-way repeated-measures ANOVA, $n = 6$/group), and MYMX (two-way repeated-measures ANOVA, $n = 5$–6/group, *$p = 0.001$). Values are group means ± SEM, each sample representing an independent animal or line of isolated primary myoblasts. * indicates a significant effect of genotype within a given condition, $p < 0.05$. Horizontal line indicates a significant effect of DM or BaCl$_2$. Source data are available online for this figure.

myogenesis. Surprisingly, however, we also determined that the function of TRIM28 is not mediated through changes in TRIM28(S473) phosphorylation. To reach this conclusion, we first performed in vitro studies with lentiviral-mediated expression of shRNA targeting *Trim28* mRNA along with "rescue" expression of wild-type or serine 473 phosphodefective TRIM28. The outcomes of this approach revealed that the shRNA-mediated loss of TRIM28 substantially inhibited the ability of primary myoblasts to fuse during myotube formation, and that this defect could be rescued by the expression of either wild-type or phosphodefective TRIM28 (Appendix Fig. S6). Combined, these results indicated that changes in TRIM28(S473) phosphorylation are not required for the role that TRIM28 plays in myoblast fusion. Numerically, the data were very compelling; however, the results starkly contrasted with observations that were previously made in C2C12 myoblasts (Singh et al, 2015). To address this discrepancy, we attempted to repeat the "rescue" studies in C2C12 myoblasts, but these experiments were confounded by a very low transduction efficiency. Importantly, with the lentiviral transduction system that was employed, the transduction efficiency could significantly alter the outcomes because it will affect the level of both the virus-encoded shRNA and protein. Thus, in addition to differences in the model systems (C2C12 myoblast vs. primary myoblast), differences in the transduction efficiency could have also led to the discrepant results. Unfortunately, the cause of the low transduction efficiency was not established but previous studies have shown that TRIM28 can repress the transcription of endogenous retroviruses (Fasching et al, 2015; Rowe et al, 2010). Since lentiviruses are a subclass of retroviruses, we suspect that the lentiviral-mediated expression of TRIM28 suppressed its own transcription and, in turn, led to low transduction efficiency (Milone and O'Doherty, 2018). Regardless of the exact cause, it became clear that a non-retroviral strategy would be needed to more clearly establish whether changes in TRIM28(S473) phosphorylation contribute to the role that TRIM28 plays in myogenesis.

When considering potential strategies, we recognized that an ideal "rescue" approach would be one that would enable us to study the functional significance of TRIM28(S473) phosphorylation specifically within the satellite cells of adult mice. Unfortunately, however, no method for such an approach had ever been described. For instance, the in vivo functional significance of a phosphorylation site is typically studied by creating mice with a knockin of a phosphodefective allele, or by using CRISPR-Cas9 to generate a phosphodefective allele (Ruvinsky et al, 2005; Saha et al, 2021). Importantly, both approaches are limited by the fact that the expression of the phosphodefective mutant is not cell-type specific and the expression is constitutive which, in our study, might lead to defects in critical processes such as

embryonic myogenesis. As such, we developed a innovative rescue-based approach that allowed for the simultaneous satellite cell-specific and tamoxifen-inducible knockout of endogenous TRIM28 and "rescue" knockin of either wild-type or phosphodefective TRIM28 (Figs. 7 and EV4). The development of this approach was important for two reasons. First, it enabled us to establish that changes in TRIM28(S473) phosphorylation do not contribute to the role that TRIM28 plays in myogenesis. Second, it provided proof-of-concept for a cell-type specific and inducible strategy that, in theory, could be used to study the functional significance of a multitude of different forms of biological regulation including genetic mutations (e.g., base substitutions, deletions, or insertions), post-transcriptional modifications (e.g., miRNA-resistant mutations), and post-translational modifications (e.g., acetylation, SUMOylation, ubiquitination, etc.). As such, we expect that our rescue-based approach will be beneficial for a wide range of fields.

Having determined that changes in TRIM28(S473) phosphorylation do not contribute to the role that TRIM28 plays in myogenesis, we wanted to gain additional insight into the mechanism(s) via which TRIM28 exerts its regulatory effect. Importantly, the results from our in vitro and in vivo models of myogenesis consistently indicated that the loss of TRIM28 does not inhibit the proliferation or differentiation of satellite cells. Instead, several lines of evidence indicated that the loss of TRIM28 leads to a defect in satellite cell fusion. For instance, we determined that TRIM28 was not required for the mechanical load-induced activation of satellite cell proliferation or differentiation, but the loss of TRIM28 did lead to a substantial decrease in myonuclear accretion which suggested that there might be an impairment at the level of satellite cell fusion (Figs. 2 and 3). Likewise, we found that TRIM28 was not required for the injury-induced activation of satellite cell proliferation or differentiation, but again, multiple lines of evidence suggested that the loss of TRIM28 led to a defect in fusion (Figs. 4 and 7; Appendix Fig. S4). Similar conclusions were also reached when we studied primary myoblasts with our in vitro myotube formation assay (Figs. 6 and EV3; Appendix Fig. S6). Given the consistent outcomes from these different model systems, we sought to gain further insight into why the loss of TRIM28 inhibits fusion, and we ultimately discovered that the loss of TRIM28 inhibits the induction of MYMX expression and fusion pore formation (Figs. 9 and 10). Intriguingly, although the loss of TRIM28 markedly inhibited the increase in MYMX that occurred during myotube formation, it did not inhibit the increase in MYMK or hemifusion. This was intriguing because it has been argued that, due to their inherent membrane-remodeling activities, stringent control of MYMK and MYMX expression is needed (Witcher et al, 2023). Thus, it bears mentioning that, during myotube formation, it has been shown that

the expression of MYMX and MYMK are both regulated by MYOD (Zhang et al, 2020). Furthermore, previous studies have shown that MYOD and TRIM28 physically interact and that TRIM28 can regulate the function of MYOD (Singh et al, 2015). As such, the potential exists for a TRIM28/MYOD-dependent mechanism that endows satellite cells with the ability to differentially regulate the expression of MYMK and MYMX. If correct, such a mechanism would enable satellite cells to stringently control the timing of MYMK and MYMX expressions and the related processes of hemifusion and concomitant fusion pore formation.

## Limitations of the study

The results of this study provide multiple lines of support for the conclusion that TRIM28 regulates myogenesis by controlling fusion pore formation. However, the exact mechanism through which TRIM28 regulates this process remains to be determined. For instance, the outcomes suggest that the regulation of fusion pore formation is due, at least in part, to the ability of TRIM28 to regulate MYMX expression. However, a role for additional fusogenic factors cannot be excluded. Moreover, the mechanism(s) via which TRIM28 regulates MYMX remain to be resolved. For instance, TRIM28 is well known for its ability to control gene expression, and as suggested above, TRIM28 might regulate the expression of MYMX at the transcriptional level (Cheng et al, 2014). However, TRIM28 could also regulate MYMX through post-translational modifications. For instance, TRIM28 possesses SUMO E3 ligase activity and this property of TRIM28 has been implicated in the regulation of the protein levels of α-Syn and tau (Rousseaux et al, 2016). TRIM28 also possesses ubiquitin E3 ligase activity which has been implicated in the regulation of the protein levels of critical factors such as p53 (Cheng et al, 2014; Jin et al, 2021). Importantly, the SUMO and ubiquitin E3 ligase activities of TRIM28 not only have the potential to regulate the protein levels of MYMX, but such modification could also regulate the spatial localization of MYMX and other fusogenic proteins (Komander, 2009; Sehat et al, 2010). Hence, there is much to learn about how TRIM28 regulates fusion pore formation, and this will be an important topic for future investigations.

## Methods

### Reagents and tools table

| Reagent/Resource | Reference or Source | Identifier or Catalog Number |
|---|---|---|
| **Experimental Models** | | |
| Wild-Type C57BL/6 Mice | Mouse Breeding Core at UW-Madison | |
| B6.129S2(SJL)-Trim28tm1.1lpc/J | Jackson Laboratories | Stock# 018552 |
| B6.Cg-Pax7tm1(cre/ERT2)Gaka/J | Jackson Laboratories | Stock# 017763 |
| TARGATT LSL-HA-hTRIM28 knockin mice | This study | |
| TARGATT LSL-HA-hTRIM28(S473A) knockin mice | This study | |
| TARGATT™ H11 mice | Applied StemCell | |
| HEK293 | ATCC | CRL-1573™ |

| Reagent/Resource | Reference or Source | Identifier or Catalog Number |
|---|---|---|
| **Recombinant DNA** | | |
| pKH3-TRIM28 | Addgene | #45569 |
| TARGATT™6.1 (CAG – L4SL – MCS – PolyA) | Applied StemCell | #AST-3050 |
| pMD2.G | Addgene | #12259 |
| psPAX2 | Addgene | #12260 |
| scrambled shRNA lentiviral plasmid | Singh et al, 2015 | |
| *Trim28* shRNA lentiviral plasmid | Singh et al, 2015 | |
| *Trim28* shRNA + shRNA resistant WT TRIM28 lentiviral plasmid | Singh et al, 2015 | |
| *Trim28* shRNA + shRNA resistant S473A TRIM28 lentiviral plasmid | Singh et al, 2015 | |
| **Antibodies** | | |
| Anti-Mouse Total Tif1β (Kap-1, TRIM28) (C42G12) | Cell Signaling Technologies | Cat# 4124S IHC: 1:30 WB: 1:1000 |
| Anti-HA-Tag (C26F4) | Cell Signaling Technologies | Cat# 3724S IHC: 1:300 |
| Anti-Mouse P-Tif1β (KAP-1, TRIM28) Ser473 Poly6446 | BioLegend | Cat# 644602 IHC: 1:30 |
| Anti-Mouse PAX7 | Developmental Studies Hybridoma Bank | AB_528428 IHC: 1:10 |
| Anti-Mouse MYH3 (embryonic myosin heavy chain) (F1.652) | Developmental Studies Hybridoma Bank | AB_528358 IHC: 1:350 |
| Anti-Mouse MYH2 (Type IIa myosin heavy chain) (SC-71) | Developmental Studies Hybridoma Bank | AB_ 2147165 IHC: 1:100 |
| Anti-Mouse MYH4 (Type IIb myosin heavy chain) (BF-F3) | Developmental Studies Hybridoma Bank | AB_ 2266724 IHC: 1:50 |
| Anti-Mouse MYH1 (Type IIx myosin heavy chain) (6H1) | Developmental Studies Hybridoma Bank | AB_2314830 IHC: 1:10 |
| Anti-Mouse MHC (all myosin heavy chain isoforms) (MF 20) | Developmental Studies Hybridoma Bank | AB_2147781 IHC: 1:50 |
| Anti-Mouse MYOD (5.8A) | BD Pharmingen | Cat# 554130 IHC: 1:50 WB: 1:1000 |
| Anti-Mouse Dystrophin (Dy8/6C5) | Novocastra | Cat# NCL-DYS2 IHC: 1:100 |
| Anti-Mouse MYOG (F5D) | Santa Cruz | Cat# sc-12732 IHC: 1:50 WB: 1:500 |
| Anti-Mouse Myomixer | ThermoFisher Scientific | Cat# AF4580SP WB: 1:2000 |
| Anti-Mouse Dystrophin | Abcam | Cat# ab15277 IHC: 1:300 |
| Anti-Mouse Myomaker | Abcam | Cat# ab188300 WB: 1:200 |

| Reagent/Resource | Reference or Source | Identifier or Catalog Number |
|---|---|---|
| Anti-Mouse Ki67 (SP6) | Abcam | Cat# ab16667 IHC: 1:50 |
| Anti BrdU (BMC9318) | Sigma-Aldrich | Cat# 11170376001 IHC: 1:20 |
| Anti-Mouse Laminin | Sigma-Aldrich | Cat# L9393 IHC: 1:500 |
| Peroxidase-labeled Anti-Sheep Secondary | Sigma-Aldrich | Cat# A3415 WB: 1:1000 |
| Fab Fragment Goat Anti-Mouse IgG (H + L) Block | Jackson Immunoresearch | Cat# 115-007-003 IHC: 1:10 |
| AMCA Anti-Mouse IgM | Jackson Immunoresearch | Cat# 115-155-075 IHC: 1:150 |
| Alexa Fluor 488 Anti-Mouse IgG1 | Jackson Immunoresearch | Cat# 115-545-205 IHC: 1:2000 |
| FITC Anti-Mouse IgG2b | Jackson Immunoresearch | Cat# 115-095-207 IHC: 1:200 |
| Alexa Fluor 594 Anti-Mouse IgG1 | Jackson Immunoresearch | Cat# 115-585-205 IHC: 1:2000 |
| Alexa Fluor 488 Anti-Rabbit IgG | Invitrogen | Cat# A11008 IHC: 1:5000 |
| Alexa Fluor 594 Anti-Rabbit IgG | Invitrogen | Cat# A11037 IHC: 1:5000 |
| Peroxidase-labeled Anti-Rabbit Secondary | Vector Labs | Cat# PI-1000 WB: 1:5000 |
| Peroxidase-labeled Anti-Mouse Secondary | Vector Labs | Cat# PI-2000 WB: 1:5000 |
| **Oligonucleotides and other sequence-based reagents** | | |
| B6.129S2(SJL)-Trim28tm1.1lpc/J Genotyping primers | Jackson Laboratories | |
| B6.Cg-Pax7tm1(cre/ERT2)Gaka/J Genotyping primers | Jackson Laboratories | |
| TARGATT LSL-HA-hTRIM28 knockin mice Genotyping primers | This study | |
| TARGATT LSL-HA-hTRIM28(S473A) knockin mice Genotyping primers | This study | |
| **Chemicals, Enzymes and other reagents** | | |
| Puromycin Dihydrochloride | ThermoFisher Scientific | Cat# 5402225MG |
| BrdU (5-Bromo-2′-deoxyuridine) | Millipore Sigma | 19-160 |
| Nilotinib | Selleck Chemical | Cat# 103538-392 |
| Sirius Red | Electron Microscopy Science | Cat# 26357-02 |
| Xylene | Electron Microscopy Science | Cat# 5029801 |
| Tamoxifen | Sigma Life Science | ID: T5648-5G |
| Phalloidin Conjugates (CF405M) | Biotium | Cat# 00034-T IHC: 1:20 |

| Reagent/Resource | Reference or Source | Identifier or Catalog Number |
|---|---|---|
| Hoechst | BD Pharmingen | Cat# 33342 IHC: 1:3000 |
| Barium Chloride Dihydrate | Sigma-Aldrich | B0750-100G |
| Ampicillin | Fisher Scientific | BP1760-25 |
| MluI | New England Biolabs | R3198S |
| NheI | New England Biolabs | R3131S |
| Bpu10I | New England Biolabs | R0649S |
| T4 DNA Ligase | New England Biolabs | M0202S |
| Matrigel | Corning | 356234 |
| Collagenase II | Worthington Biochemical | 9001-12-1 |
| VybrantTM DiD | Thermo Fisher Scientific | V22887 |
| Green CMFDA cell tracker | Thermo Fisher Scientific | C7025 |
| Deep Red cell tracker | Thermo Fisher Scientific | C34565 |
| **Software** | | |
| Nikon NIS-Elements D software | Nikon | |
| PRISM | GraphPad Software | |
| SigmaPlot | Systat Software | |
| ImageJ | National Institutes of Health | |
| CellProfiler | BROAD Institute | |
| **Other** | | |
| BZ-X700 Keyence microscope | Keyence | |
| Nikon 80i epifluorescence microscope | Nikon | |
| Leica TCS SP8 confocal laser scanning microscope | Leica | |
| 809C in situ mouse apparatus | Aurora | |
| UVP Autochemi system | Analytika Jena | |
| No-StainTM Protein Labeling reagent | Thermo Fisher Scientific | A44717 |
| QuickTiterTM Lentivirus Titer Kit | Cell Biolabs | VPK-107 |
| In Situ Cell Death Detection Kit, Fluorescein | Roche | 11684795910 |

## Animals

Male and female C57BL6 mice were obtained from the University of Wisconsin-Madison Mouse Breeding Core. Tamoxifen-inducible and satellite cell-specific *Trim28* knockout C57BL6 mice were generated by crossing *Trim28*$^{flox/flox}$ (Jackson Laboratory, Stock# 018552) with *Pax7*-iCre$^{+/-}$ (Jackson Laboratory, Stock# 017763) mice. Genotyping protocols for tail DNA of these mice can be found on the Jackson Laboratory website. The TARGATT system was used to create a rescue mouse that allowed for simultaneous satellite cell-specific and tamoxifen-inducible knockout of endogenous TRIM28 and knockin "rescue" expression of HA-tagged

human TRIM28 (HA-hTRIM28). Specifically, HA-hTRIM28 was removed from pKH3-TRIM28 plasmid DNA (Addgene, #45569) via digestion with NheI and MluI in CutSmart buffer (New England Biolabs). The HA-hTRIM28 construct was then ligated into the multiple cloning site (MCS) of TARGATT™6.1 (CAG – L4SL – MCS – PolyA) vector which contains a LoxP-StopP-LoxP (LSL) cassette and an attB integration sequence (Applied StemCell, #AST-3050). Embryos from TARGATT™ H11 mice (Applied StemCell) which contained an attP integration site in the H11 locus were injected with ΦC31 integrase plus TARGATT™6.1-LSL-HA-hTRIM28 (Chen-Tsai, 2019). The offspring with successful integration were crossed with the *Pax7*-iCre$^{+/-}$ : *Trim28*$^{flox/flox}$ mice to generate (KO + hTRIM28) mice with the genotype *Pax7*-iCre$^{+/-}$ : *Trim28*$^{flox/flox}$ : LSL-HA-hTRIM28$^{+/+}$. CRISPR-Cas9-mediated homologous-directed repair was used to make a single point mutation in the hTRIM28 that switched the serine 473 residue to a non-phosphorylatable alanine. Specifically, a sgRNA (GGTGTGAAACG GTCCCGCTC) was designed to introduce a double-stranded break at the region corresponding to human Trim28 exon 12. A single-stranded oligodeoxynucleotide (ssODN) homology-directed repair (HDR) donor template containing a T1416 to G1416 mutation with 75 5′ nucleotides and 76 3′ nucleotide homology arms was used to introduce the mutation through HDR: TGAGGTCCAGGTCCAGGCGTTCAAGGCTCAC TCGTGGCACCTTGCGCATAAGGCCGCTCACCTCGCCCTCA CCTGcGCGGGACCGTTTCACACCTGACACATGGGGCTCTG-CACTTGAGTAGGGATCATCTCCTGACCCAAAGCCATAG CCT. The ssODN HDR template and Cas9/sgRNA RNP were co-electroporated into the embryos of the TARGATT LSL-HA-hTRIM28 knockin mice. Successfully edited offspring were crossed with the *Pax7*-iCre$^{+/-}$: *Trim28*$^{flox/flox}$ mice to generate (KO + hTRIM28(S473A)) mice with the genotype of *Pax7*-iCre$^{+/-}$: *Trim28*$^{flox/flox}$: LSL-HA-hTRIM28(S473A)$^{+/+}$. PCR with primers *p1* (5′ TCTACTGGAGGAGGACAA ACTGGTCAC), *p2* (5′ GAC-GATGT AGGTCACGGTCTCGAAG) and *p3* (5′ TGACACA TCCTGCCCTTACCTTACTACC), were used to confirm the knockin of hTRIM28 from tail DNA as illustrated in Fig. EV4A,B. PCR with primers *p4* (5′ ATTTCAGTGG GACCTCAATGCC) and *p5* (5′ GTGA CAGTCCAGGTGGAAACAAA) and Bpu10I digestion (Thermo Fisher Scientific #FD1184) was used to confirm the genomic differences between the KO + hTRIM28 and the KO + hTRIM28(S473A) mice as shown in Fig. EV4C,D. PCR products of F0 and F1 founders were also sequenced through amplicon sequencing followed with CRISPResso2 analysis to confirm the mutation.

All mice were given food and water ad libitum and kept in a room maintained at 25 °C with a 12-h light: dark cycle. Genotypes were confirmed with PCR of tail snips when the mice were 3 weeks of age and, where indicated, 6-week-old mice were given daily intraperitoneal injections of 100 mg/kg of tamoxifen for five consecutive days as previously described (You et al, 2019). All experiments, except Fig. 1A–D (all male) were designed in a sex-balanced manner, and interventions were performed when the mice were 8 to 10 weeks of age. For all surgical interventions mice were anesthetized with 1–5% isoflurane mixed in oxygen and given an intraperitoneal injection of 0.05 μg/g of buprenorphine to relieve any procedure-induced pain. Euthanasia was performed with cervical dislocation while the mice were under isoflurane anesthesia. All animal experiments were approved by the Institutional

Animal Care and Use Committee of the University of Wisconsin-Madison (#V005375). All procedures were in compliance with ethical regulations.

## Maximal-intensity contractions

Maximal-intensity contractions were performed in anesthetized male C57BL6 mice as previously described (Steinert et al, 2021). In brief, the sciatic nerve of the right thigh was exposed by a small incision along the lateral edge. An electrode was then placed on the sciatic nerve and contractions were elicited by stimulating the nerve with an SD9E Grass stimulator (Grass Instruments) at 100 Hz with 0.5 ms pulses at 4–7 V for a total of 10 sets of 6 contractions. Each contraction lasted for 3 s and was followed by a 10-s rest period. A 1 min rest period was provided between each set. At 1 h after the last contraction, both the right and left TA muscles were collected and immediately processed for immunohistochemistry as detailed below.

## Synergist ablation

As previously described (Goodman et al, 2011), mice were anesthetized and subjected to unilateral synergist ablation by removing the soleus and distal two-thirds of the gastrocnemius muscle. A sham surgery was performed on the contralateral hindlimb. The plantaris muscles were collected at 4 or 14 days post-surgery for immunohis-tochemical analysis as detailed below. As indicated in the legend for the experiments in Fig. 3, some of the mice were also given daily intraperitoneal injections of 50 μg/g BrdU (Sigma-Aldrich) that was dissolved in PBS at a concentration of 5 mg/mL.

## BaCl₂-induced muscle injury

The right TA muscle of anesthetized mice was injected at the proximal and distal ends with 25 μL of 1.2% $BaCl_2$ (50 μL in total), while the left TA muscle was injected in the same manner with PBS as a control condition. The TA muscles were collected at the indicated time points for immunohistochemistry or western blot analysis as detailed below. For the experiments in Fig. EV2C–F, the mice also received daily intraperitoneal injections of 20 mg/kg Nilotinib (Selleck Chemical) dissolved in DMSO (Thermo Fisher Scientific) at a concentration of 5 mg/mL, or 4 mL/kg DMSO as a control condition, on days 3–5 following muscle injury.

## In situ muscle contraction

Mice were anesthetized and the TA muscle was subjected to in situ measurement of force production as previously described (Steinert et al, 2021). In brief, single-pulse stimulations (60 V for 0.5 ms) were used to determine the optimal length of the muscle and then the muscles were subjected to 300 ms tetanic contractions at 100, 150, and 200 Hz with 2 min between each stimulation. The highest force produced during these contractions was recorded as the peak tetanic force.

## Lentivirus production

Lentiviral plasmids encoding scrambled shRNA (CNT), shRNA targeting *Trim28* mRNA (KD), shRNA targeting *Trim28* mRNA along with expression of a shRNA resistant form of TRIM28

(KD + WT), or shRNA targeting *Trim28* mRNA along with expression of an shRNA resistant form of phosphodefective TRIM28 (KD + PD) in which the serine 473 residue had been mutated to a non-phosphorylatable alanine. These lentiviral plasmids have been previously described (Singh et al, 2015) and were kindly provided by Dr. Didier Trono (École Polytechnique Fédérale de Lausanne). The lentiviral plasmids were propagated in DH5α E. coli or Stbl3 E. coli with LB broth containing 60 μg/ml Ampicillin, and purified with an Endofree plasmid kit.

HEK293 (RRID:CVCL_0045) cells were grown to 90% confluence in 10 cm plates (Corning) with DMEM (Hyclone) containing 10% fetal bovine serum (FBS) (Invitrogen). The cells were co-transfected with 2 μg lentivirus packaging plasmids (pMD2.G (Addgene #12259) and 10 μg psPAX2 (Addgene #12260)) along with 10 μg lentiviral plasmid with lipofectamine 3000 (Thermo Fisher Scientific). After 15 h, the medium containing viral plasmids was replaced with DMEM + 10% FBS and 1% penicillin/streptomycin (Hyclone), and the cells were cultured for another 72 h. After 72 h, the viral supernatant was harvested, and the virus titer was measured based on the titer of the viral protein p24 with QuickTiter™ Lentivirus Titer Kit (Cell Biolabs) following the manufacturer's instructions.

## Primary myoblasts

To create multiple primary myoblast lines as biological replicates, we isolated primary myoblasts from different mice with the following conditions. Primary myoblast lines used in Fig. EV3 were isolated from four pairs of WT and KO mice (sex balanced) at 14 days post tamoxifen injection. Primary myoblast lines used for the TRIM28 knockdown experiments in Figs. 6, 9, 10 and Appendix Fig. S6 were isolated from four 6-week-old C57BL/6J mice (sex balanced) and maintained following the protocol published by Hindi et al (Hindi et al, 2017). Specifically, hindlimb muscles were harvested and minced in sterile PBS containing 1% penicillin/streptomycin. The minced muscles were centrifuged at $15,000 \times g$ for 1 min and transferred to 10 ml of digestion buffer (DMEM containing 2.5% HEPES, 1% penicillin/streptomycin, and 400 U/ml collagenase II (Worthington Biochemical)) and placed on an orbital shaker at 100 rpm for 1 h at 37 °C. Digested tissues were centrifuged at $1500 \times g$ for 5 min, and the resulting pellets were resuspended in neutralizing medium (DMEM containing 10% FBS and 1% penicillin/streptomycin). Resuspended cell mixtures were filtered through a 70 μm strainer (Thermo Fisher Scientific) and then a 30 μm strainer (Miltenyi Biotec). The filtered cell mixtures were spun at $1500 \times g$ for 5 min and resuspended in myoblast growth medium (GM) (F-10 medium (Thermo Fisher Scientific) containing 20% FBS, 10 ng/ml basic fibroblast growth factor (Peprotech), and 1% penicillin/streptomycin). During the isolation procedure, 10 cm dishes were coated with 3 ml of 10% Matrigel (Corning) in DMEM for 1 min. Excess Matrigel was removed and then the plates were air dried for 20 min. Resuspended cell mixtures were seeded onto the Matrigel-coated dishes (the entire cell mixture from 1 animal per 10 cm dish) and grown in 5% $CO_2$ at 37 °C. At 72 h post-seeding, adherent primary myoblasts and large triangular-shaped fibroblasts were present. The GM was discarded to remove any unattached cells and the remaining cells were gently washed with F-10 medium containing 1% penicillin/streptomycin. Adherent cells were then detached by treatment with 0.25% trypsin

(Mediatech) in PBS. The detached cell mixture was neutralized with GM and transferred to a regular 10 cm dish for 45 min. During this incubation, larger non-myoblast cells (e.g., fibroblasts) adhered to the bottom of the dish while the majority of the primary myoblasts remained suspended in the media. The unattached cell mixture was then seeded on another Matrigel pre-coated dish and propagated. The cells were grown to 50–60% confluence and then the detachment plus pre-plating procedure for removing non-myoblasts was repeated before further propagation. The procedure for removing fibroblasts was repeated for 2–3 passages so that a myoblast purity of ~95% was obtained. After the cell mixture was enriched by the first pre-plating procedure, the cells were defined as passage 1. The primary myoblasts used for knockdown and rescue-based (Figs. 6, 9, 10 and Appendix Fig. S6) were between passages 6 and 10. For the primary myoblasts that were isolated from tamoxifen-injected WT and KO mice (Fig. EV3), passage 2 cells were used for experimentation. Thus, the pre-plating procedure was only completed 2 times, and the myoblast purity in these experiments was ~80%. To account for this, care was taken to ensure that fibroblasts (defined as large mononucleated triangular-shaped cells with large nuclei) were not included during the quantification of differentiation and fusion indices. In all cases, propagating primary myoblasts were always maintained below 70% confluence to prevent differentiation.

For primary myoblast knockdown and resue-based experiments, $2.5 \times 10^5$ WT primary myoblasts were seeded per well on 6-well plates (Corning Costar). The next day, the primary myoblasts were infected at a multiplicity of infection (MOI) of 100 with WT, KD, KD + WT, or KD + PD lentivirus in GM containing 8 μg/ml polybrene (EMD Millipore). The virus-containing GM was replaced with regular GM at 16 h post-infection. At 2–3 days post-infection (2 days for knockdown experiments (Figs. 6, 9, and 10), and 3 days for rescue-based experiments due to low transduction efficiency (Appendix Fig. S6)), the primary myoblasts were selected with 2 μg/ml puromycin in GM for 72 h. After selection, the surviving primary myoblasts were subjected to the experiments described in Figs. 6, 9,10 and Appendix Fig. S6.

## Myotube formation assay

A total of $1.5 \times 10^5$ or $3 \times 10^5$ primary myoblasts in GM were plated per well of 24-well or 6-well plates (Corning Costar) that had been pre-coated with 10% Matrigel, respectively. At 1 day post-seeding for 24-well plates, or 2 days post-seeding for 6-well plates, the GM was replaced with differentiation medium (DM) (DMEM containing 2% horse serum (Thermo Fisher Scientific) and 1% penicillin/streptomycin). If lentivirus-infected myoblasts were used, 2 μg/ml puromycin (Thermo Fisher Scientific) was added to the GM and DM. After 48 h in DM, the myoblasts/myotubes were subjected to immunohistochemistry as described below.

## Intracellular content and lipid membrane probe labeling

CNT and KD primary myoblasts that had undergone 24 h of differentiation were detached with 0.25% trypsin and resuspended in GM to neutralize trypsin. Then, the cells were washed with F-10 medium 3 times to deplete FBS and finally resuspended in FBS-free F-10 medium at a density of $10^6$ cells/ml. Cells were then incubated with 5 μM of the fluorescent lipid probe Vybrant™ DiD (Thermo

Fisher Scientific) or 10 µM of the intracellular content probe Green CMFDA cell tracker or Deep Red cell tracker (Thermo Fisher Scientific). Cells were incubated with the probe for 20 min at 37 °C to allow for labeling. The labeled cells were then washed with GM 3 times and resuspended in DM. Different combinations of the probe-labeled cells were co-plated at a 1 to 1 ratio ($3.0 \times 10^5$ cells/well of a 24-well plate) as indicated in Fig. 7. The co-plated cells were maintained in DM for an additional 24–30 h and then subjected to immunohistochemistry as described below.

## Antibodies

All antibodies and the dilutions that were employed are listed in the Reagents and Tools table as well as in Table EV1 for a quick overview.

## Western blot analysis

Primary myoblasts were lysed with WIK buffer [(40 mM Tris(pH 7.5), 1 mM EDTA, 5 mM EGTA, 0.5% Triton X-100, 25 mM β-glycerophosphate, 25 mM NaF, 1 mM $Na_3VO_4$, 10 mg/ml leupeptin, and 1 mM PMSF], and the lysates were centrifuged at $500 \times g$ for 5 min at 4 °C. Frozen muscle tissues were homogenized with a Polytron for 30 s in ice-cold WIK buffer as previously described (Steinert et al, 2021). A DC protein assay kit (BioRad) was used to determine the protein concentration of individual samples. Equal amounts of protein from each sample were subjected to SDS-PAGE and transferred to a PVDF membrane at 300 mA for 1 h and 45 min. Immediately after the transfer step was complete, total protein loading on the PVDF membrane was stained with No-Stain™ Protein Labeling reagent (Thermo Fisher Scientific) following the manufacturer's instructions. The image acquired from this staining procedure was used to quantify the total amount of protein transferred per sample. The PVDF membrane was then further blocked with 5% milk in TBST, and incubated with primary and secondary antibodies as previously described (You et al, 2019). Of note, the myomaker antibody was pre-incubated with a PVDF membrane at 4 °C overnight to eliminate non-specific binding. For visualization, membranes were incubated with ECL-prime reagent (Amersham) and imaged with a UVP Autochemi system (Analytika Jena). All western blots were quantified with ImageJ software (NIH).

## Immunohistochemistry

Freshly harvested muscles were embedded in optimal cutting temperature compound (Sakura Tissue-Tek) and frozen in liquid nitrogen-chilled isopentane for 30 s. A cryostat at −20 °C was then used to obtain 10 µm thick mid-belly cross-sections. For conditions in which P-TRIM28(S473), or Type IIa, IIx, or IIb MHC and laminin were analyzed, the sections were fixed for 10 min in −20 °C acetone. For all other analyses, the sections were fixed at room temperature (RT) for 10 min in 1% paraformaldehyde (PFA) in PBS. For studies involving the analysis of myoblasts/myotubes, the cells were fixed with 4% PFA in PBS for 10 min at RT. Fixed cells/sections were washed with PBS and blocked with buffer A (0.5% Triton X-100 and 0.5% bovine serum albumin (BSA) in PBS) for 20 min at RT. For most conditions, the blocked sections were incubated with primary antibodies or Phalloidin Conjugates

(Biotium) dissolved in buffer A for 1 h at RT. However, for the analysis of Pax7 and MyoG incubations were performed overnight at 4 °C. If additional primary antibodies were being used in conjunction with Pax7 or MyoG, these incubations were performed the following day for 1 h at RT. After the incubation with primary antibodies, the sections were washed with PBS and then incubated with secondary antibodies in buffer A for one hour at RT. Sections were again washed with PBS, and when indicated, were incubated with Hoechst (BD Bioscience) for 5 min at RT and subjected to additional washes with PBS. The stained sections were mounted in a layer of ProLong Gold anti-fade mounting medium (Invitrogen), overlaid with a coverslip (Thermo Fisher Scientific).

In addition to the procedures described above, the protocols that involved the detection of Pax7, MYOG, or MYOD also required the use of an antigen retrieval procedure. Specifically, immediately after the PFA fixation, the sections were washed with PBS and permeabilized with 0.2% Triton X-100 in PBS for 10 min at RT. The sections were then incubated with 10 mM sodium citrate pH 6.5 for 40 min (10 min at 65 °C followed by 30 min of cooling at RT). Following the antigen retrieval procedure the sections were blocked with buffer A and subjected to the remainder of the workflow described above.

For the analysis of BrdU incorporation, fixed sections were denatured with buffer B (2 N HCl plus 0.5% Triton X-100 in PBS) for 1 h at RT and then neutralized with TBS buffer (20 mM Tris and 150 mM NaCl in water) pH 8.4 for 10 min at RT. The neutralized sections were then blocked with buffer A and subjected to the remainder of the workflow described above.

## Sirius Red staining

Mid-belly cross-sections from TA muscles were fixed with 4% paraformaldehyde (PFA) in PBS for 10 min. The fixed sections were washed with distilled water and then incubated with a 0.1% Sirius Red solution (Electron Microscopy Science) for 1 h at RT. Stained sections were washed twice with acidified water (0.5% acetic acid) for 2 min per wash. The sections were then dehydrated with 2 washes of 95% ethanol followed by another 2 additional washes of 100% ethanol (2 min per wash). Dehydrated sections were mounted in cytoseal™ XYL (Thermo Fisher Scientific).

## TUNEL assay

Mid-belly cross-sections from TA muscles that were collected 10 days following $BaCl_2$ injection were fixed with 4% paraformaldehyde (PFA) in PBS for 20 min. The sections were washed with PBS and then permeabilized with 0.1% Triton X-100 in 0.1% sodium citrate for 2 min at 4 °C. The sections were again washed with PBS and then incubated with 0.3% hydrogen peroxide in methanol for 30 min at room temperature. The sections were washed with PBS, as much liquid as possible was removed from the slide, and then 50 µL of the TUNEL reaction mixture (In Situ Cell Death Detection Kit, Fluorescein, Roche) was added to each slide. Sections were incubated for 1 h in a humid chamber, protected from light then washed with PBS. Once this TUNEL assay was completed the sections were subjected to additional immunohistochemical staining, as outlined above.

## Image analysis and quantification

For myofiber CSA measurements of the plantaris muscle, whole muscle cross-section images were captured with a Nikon 80i epifluorescence microscope. For each muscle cross-section, the CSA of at least 70 randomly selected type IIa, IIx, and IIb myofibers were measured by tracing the laminin border with Nikon NIS-Elements D software (Nikon) as previously described (Steinert et al, 2021). The mean CSA of measured type IIa, IIx, and IIb myofibers was defined as the type II myofiber CSA. For myofiber CSA measurements in the TA muscles, images of the whole muscle cross-sections were captured with a BZ-X700 Keyence microscope and then measurements were determined using a custom CellProfiler pipeline, as previously described (Zhu et al, 2021).

Myonuclei, interstitial nuclei, and BrdU-positive myonuclei were identified by capturing 3–5 randomly selected fields within each muscle cross-section with an inverted Leica TCS SP8 confocal laser scanning microscope. P-TRIM28(S473) positive nuclei, BrdU-positive myonuclei, and total myonuclei were manually quantified by using Leica LASX software (Leica). Myonuclei were defined as nuclei that resided inside of the dystrophin boundary of individual myofibers, whereas interstitial nuclei were defined as nuclei that reside outside of the dystrophin boundary of the myofibers.

To determine the percentage of myoblasts/myofibers that were at the stage of fusing, fused, mature, or pre-existing, 3–5 randomly selected images from each muscle cross-section were captured with a Nikon 80i epifluorescence microscope and then each myoblast/myofiber in the image was categorized as described in Fig. 4B–E.

For the quantification of Sirius Red stained sections, images of whole muscle cross-sections were captured with a BZ-X700 Keyence microscope. Images were imported into ImageJ and the border of cross-sections were manually traced to clear the region outside of the cross-sections with the "Clear Outside" function. The trimmed images were then separated into red, green, and blue channels. In the red channel, the intensity of the Sirius Red signal within the myofibers was set as the background in the threshold intensity program of ImageJ. The area occupied by Sirius Red (Sirius Red signal above background intensity) was then measured and used to calculate the percent of the total muscle CSA that was occupied by Sirius Red.

All other immunohistochemistry image quantifications were completed by first capturing 3–5 randomly selected fields per sample with a Nikon 80i epifluorescence microscope. The images were assessed for the elements of interest. Importantly, all quantitative analyses were performed by investigators who were blinded to the sample identification.

## Statistical analysis

Statistical significance was determined by using the Student's t-test (independent or paired), one-way ANOVA, two-way ANOVA, or two-way repeated-measures ANOVA with Student-Newman-Keuls post hoc analysis, or simple linear regression analysis, as indicated in each figure legend. An a priori alpha level for statistical significance was set at 0.05. Each experiment was repeated multiple times, with the number of independent samples being specified by the $n$ values in the figure legend. All data sets except for Fig. 1A–E (all male) were created with an equal number of male and female samples. In data sets with $n \geq 6$ samples, individual values that resided more than 3 standard deviations away from the mean were considered to be outliers and excluded from the final analyses. All statistical analyses were performed with Prism (GraphPad Software, San Diego, CA, USA) or SigmaPlot software (Systat Software, San Jose, CA, USA).

## Data availability

This study includes no data deposited in external repositories. The source data for the microscopy images included in this manuscript can be found in the BioImage Archive under the accession number S-BIAD1194.

The source data of this paper are collected in the following database record: biostudies:S-SCDT-10_1038-S44319-024-00227-1.

## Peer review information

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

## Acknowledgements

We would like to thank Drs. Marco Cassano and Didier Trono for providing plasmids that were used for lentiviral production. We also would like to thank Dr. Dustin Rubinstein and Kathy Krentz at the Genome Editing and Animal Models Facility (University of Wisconsin-Madison) for their assistance with the generation of the knockin mice. The research reported in this publication was supported by the National Institute of Arthritis and Musculoskeletal and Skin Diseases of the National Institutes of Health under Awards AR074932 to TAH, AR074932-S1 to TAH and JEH, and P30AR066524 to VS.

## Author contributions

**Kuan-Hung Lin**: Conceptualization; Formal analysis; Investigation; Visualization; Methodology; Writing—original draft; Writing—review and editing. **Jamie E Hibbert**: Conceptualization; Formal analysis; Funding acquisition; Investigation; Visualization; Methodology; Writing—original draft; Writing—review and editing. **Corey GK Flynn**: Investigation; Visualization. **Jake L Lemens**: Investigation; Writing—review and editing. **Melissa M Torbey**: Investigation; Writing—review and editing. **Nathaniel D Steinert**: Conceptualization; Investigation; Methodology. **Philip M Flejsierowicz**: Investigation; Writing—review and editing. **Kiley M Melka**: Investigation; Writing—review and editing. **Garrison T Lindley**: Investigation. **Marcos Lares**: Investigation. **Vijayasaradhi Setaluri**: Investigation. **Amy J Wagers**: Investigation. **Troy A Hornberger**: Conceptualization; Formal analysis; Supervision; Funding acquisition; Investigation; Methodology; Writing—original draft; Writing—review and editing.

Source data underlying figure panels in this paper may have individual authorship assigned. Where available, figure panel/source data authorship is listed in the following database record: biostudies:S-SCDT-10_1038-S44319-024-00227-1.

## Disclosure and competing interests statement

The authors declare no competing interests.

# Expanded View Figures

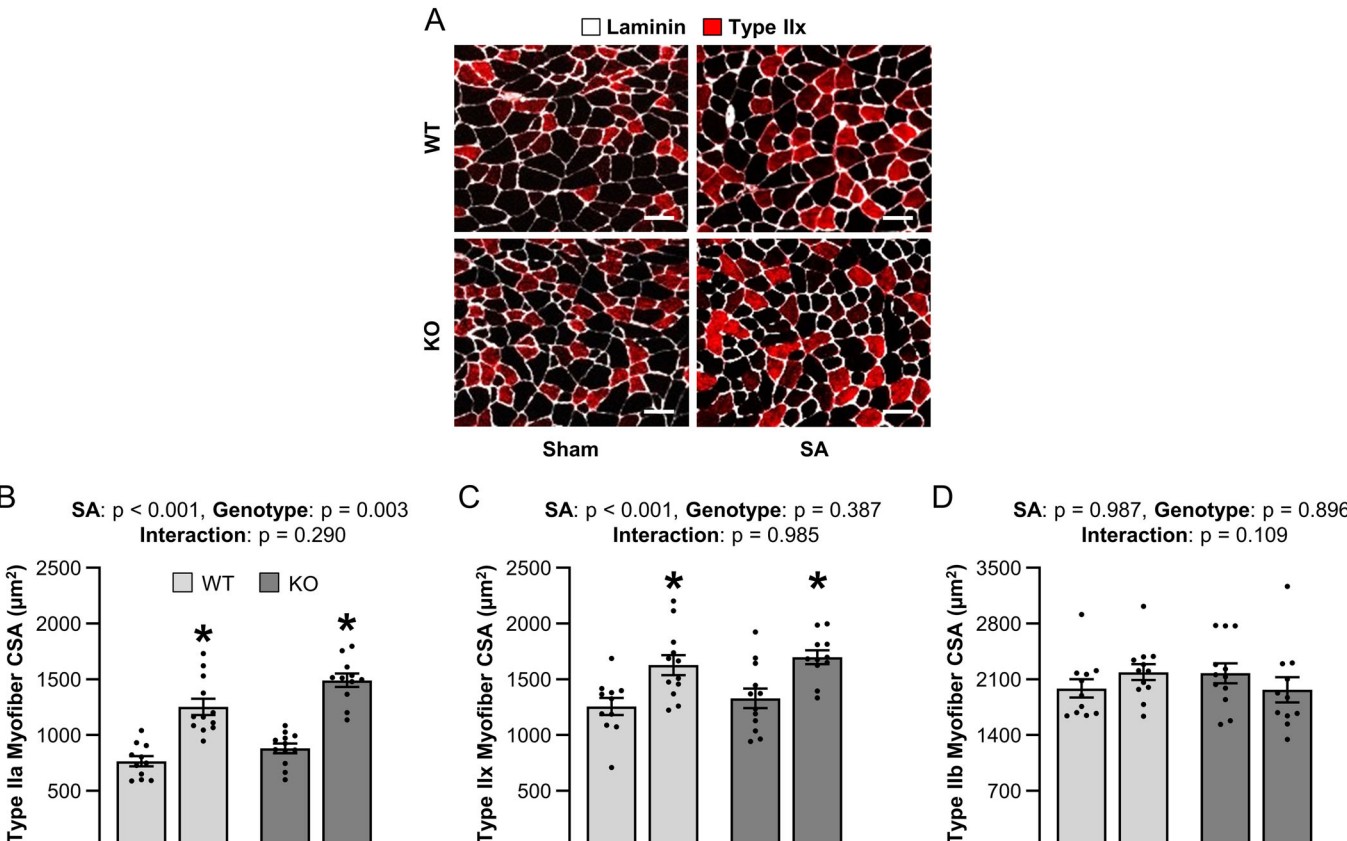

**Figure EV1.  The loss of TRIM28 in satellite cells does not impact the mechanical load-induced increase in the size of Type IIa or IIx myofibers.**

Wild-type (WT) mice and tamoxifen-inducible satellite cell-specific TRIM28 knockout mice (KO) mice were treated with tamoxifen. At 14 days post tamoxifen, mice were subjected to unilateral synergist ablation surgery (SA+), with the non-ablated limb serving as a sham control (SA-). The mice were treated as described in Fig. 3 and the plantaris muscles were collected at 14 days after the SA surgery. (**A**) Mid-belly cross-sections were subjected to immunohistochemistry for laminin and type IIx myofibers. (**B–D**) Quantification of the type IIa (two-way ANOVA, $n = 11–12$/group, $*p < 0.001$), type IIx (two-way ANOVA, $n = 11–12$/group, $*p = 0.023$ or 0.0025), and type IIb (two-way ANOVA, $n = 11–12$/group) myofiber cross-sectional area, respectively. Values are group means + SEM. * indicates a significant effect of SA within the given genotype, $p < 0.05$. Scale bars = 50 µm.

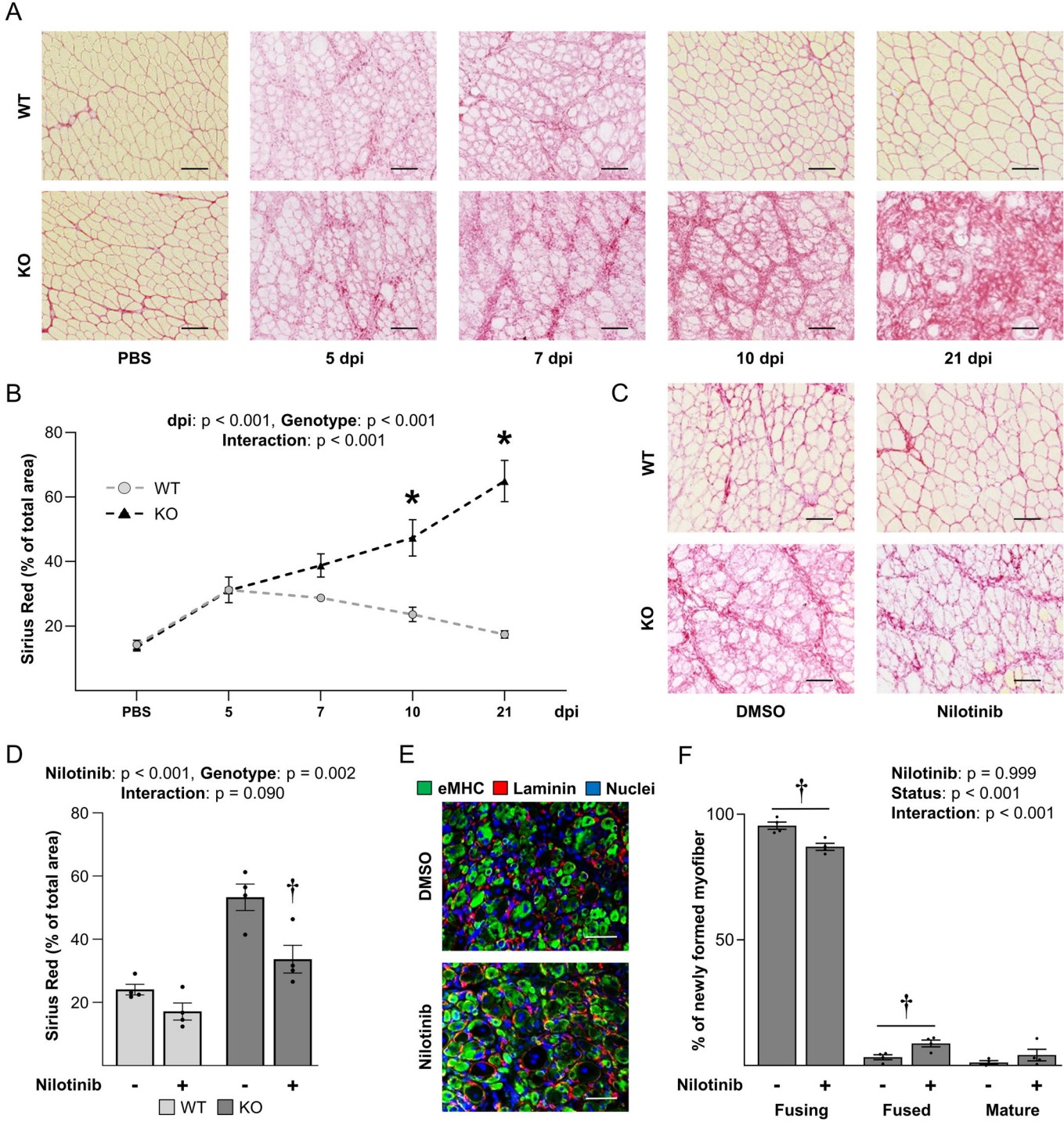

**A**

WT / KO

PBS | 5 dpi | 7 dpi | 10 dpi | 21 dpi

**B**

dpi: p < 0.001, Genotype: p < 0.001
Interaction: p < 0.001

Sirius Red (% of total area)

○ WT
▲ KO

PBS 5 7 10 21 dpi

**C**

WT / KO

DMSO | Nilotinib

**D**

Nilotinib: p < 0.001, Genotype: p = 0.002
Interaction: p = 0.090

Sirius Red (% of total area)

Nilotinib  −  +  −  +

☐ WT  ☐ KO

**E**

■ eMHC  ■ Laminin  ■ Nuclei

DMSO

Nilotinib

**F**

Nilotinib: p = 0.999
Status: p < 0.001
Interaction: p < 0.001

% of newly formed myofiber

Nilotinib  −  +  −  +  −  +
Fusing    Fused    Mature

◀ **Figure EV2. The loss of TRIM28 in satellite cells leads to excessive fibrosis following BaCl2-induced injury, but reducing the fibrosis only minimally improves the impairment in fusion.**

Wild-type (WT) mice and tamoxifen-inducible satellite cell-specific TRIM28 knockout mice (KO) mice were treated with tamoxifen. At 14 days post tamoxifen, their tibialis anterior (TA) muscles were injected with BaCl2 to induce injury or PBS as a control condition. (A) At 5, 7, 10, and 21 days post-injury (dpi), TA muscles were collected and mid-belly cross-sections were subjected to Sirius Red staining as a marker of collagen deposition. (B) Proportion of the muscle cross-section that stained positive for Sirius Red in (A) (two-way ANOVA, $n = 4$–6/group, *$p = 0.0001$ or <0.0001). (C–F) Daily intraperitoneal injections of Nilotinib (20 mg/kg/day) or DMSO were administered at 3 to 7 dpi. (C) TA muscles were collected at 10 dpi and mid-belly cross-sections were subjected to Sirius Red staining. (D) Proportion of the muscle cross-sections that stained positive for Sirius Red in (C) (two-way ANOVA, $n = 4$/group, *$p < 0.0001$, †$p = 0.0016$). (E) Mid-belly cross-sections of the TA muscles at 10 dpi were subjected to immunohistochemistry for eMHC, laminin, and nuclei. (F) Proportion of myoblasts/myofibers that were at the stage of fusing, fused, or mature as described in Fig. 4 (two-way ANOVA, $n = 4$/group, †$p = 0.0006$ or 0.0145). Values are group means + SEM. * indicates a significant difference between genotypes at the given condition, † significant effect of Nilotinib within the given genotype or stage $p < 0.05$. Scale bars = 100 μm in (A) and (C), and 50 μm in (E).

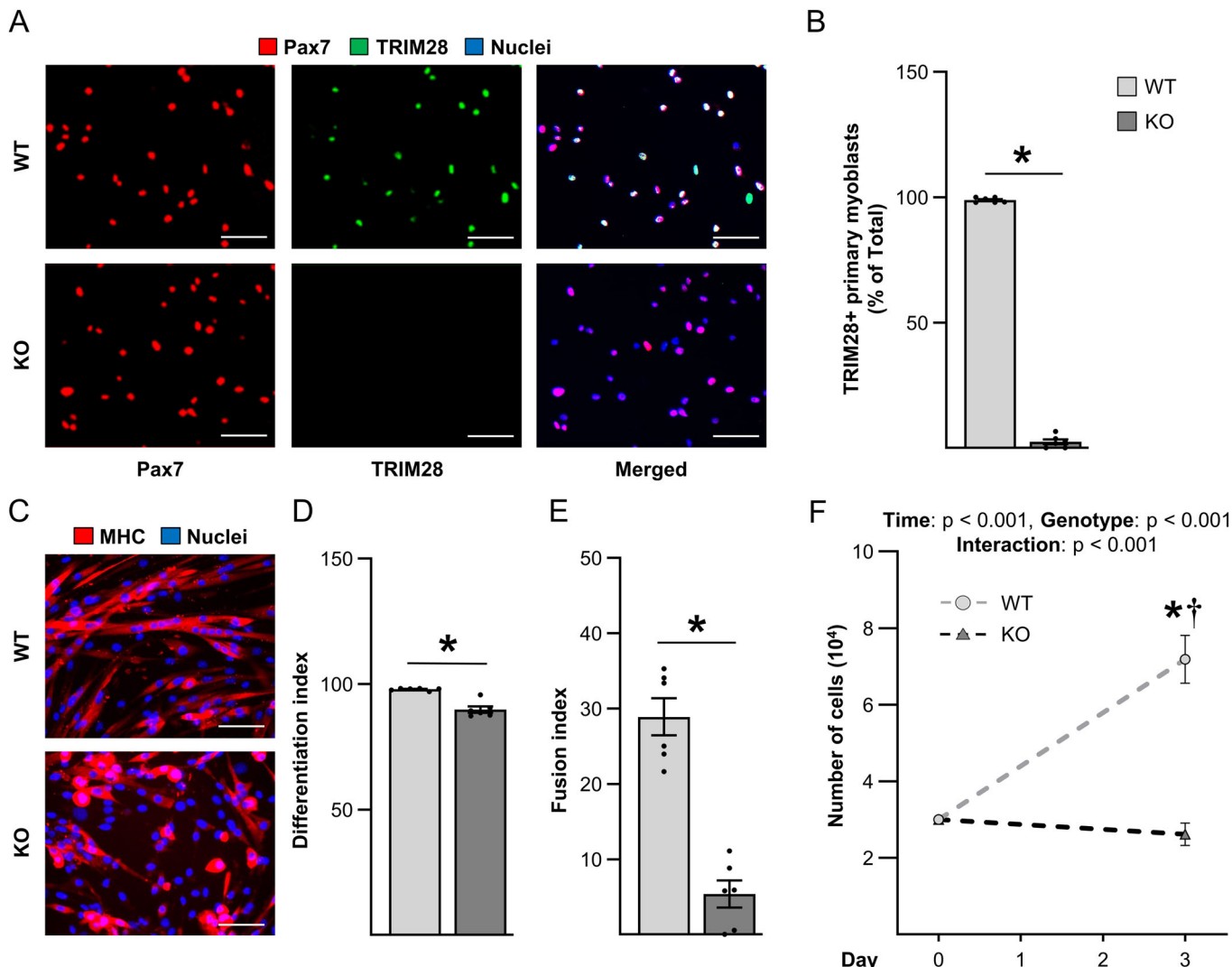

**Figure EV3. The loss of TRIM28 in satellite cells leads to a fusion defect during in vitro myotube formation.**

Wild-type (WT) mice and tamoxifen-inducible satellite cell-specific TRIM28 knockout mice (KO) mice were treated with tamoxifen. (A) At 14 days post tamoxifen, primary myoblasts were isolated, cultured in growth medium, and then subjected to immunohistochemistry for Pax7, TRIM28, and nuclei. (B) The proportion of the Pax7 positive primary myoblasts that expressed TRIM28 in (A) (unpaired Student's t-test, $n = 6$/group, *$p < 0.0001$). (C–E) WT and KO primary myoblasts were subjected to a myotube formation assay and immunohistochemistry for myosin heavy chain (MHC) and nuclei. (D) The differentiation index (% of nuclei inside MHC positive cells) (unpaired Student's t-test, $n = 6$/group, *$p < 0.0001$), and (E) the fusion index (% of nuclei inside MHC positive multinucleated cells) (unpaired Student's t-test, $n = 6$/group, *$p < 0.0001$) were quantified. (F) $3 \times 10^4$ primary myoblasts were seeded on day 0 and cultured in growth medium. The number of primary myoblasts was quantified on day 3 (two-way repeated-measures ANOVA, $n = 6$/group, * and † $p < 0.0001$). Values are group means + SEM, each sample representing an independent line of isolated primary myoblasts. * indicates a significant difference between genotypes within a given condition, † indicates a significant difference from day 0, $p < 0.05$. Scale bars = 50 µm.

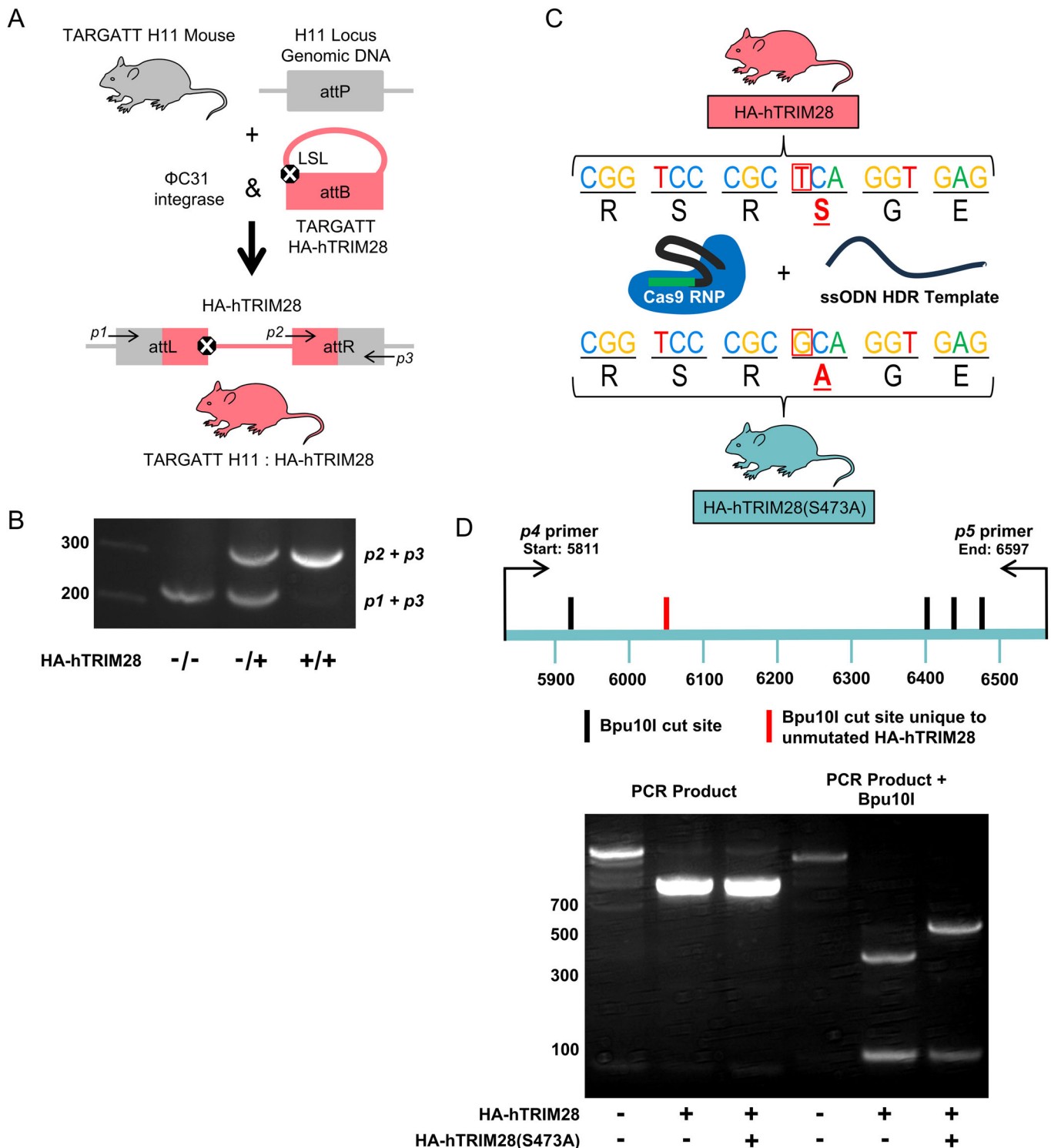

**Figure EV4.  Strategy for creating mice that allow for the tamoxifen-inducible expression of human TRIM28 or a S473A phosphodefective mutant of hTRIM28.**

(A) Embryos from TARGATT mice that contained an attP integration site in the H11 locus were injected with ΦC31 integrase and an HA-hTRIM28 TARGATT vector that contained a LoxP-Stop-LoxP (LSL) cassette and an attB integration sequence. (B) As illustrated in A, three primers (*p1, p2,* and *p3*) were used to confirm the successful integration of the HA-hTRIM28 vector into the genomic DNA of the offspring. (C) CRISPR-Cas9-mediated homologous-directed repair was used to make a single point mutation in TARGATT H11 : HA-hTRIM28 mice that switched the serine 473 residue of hTRIM28 to a non-phosphorylatable alanine (HA-hTRIM28(S473A)). (D) Primers *p4* and *p5* along with Bpu10I digestion were used to confirm the differences in the genomic DNA of the mice that expressed HA-hTRIM28 versus HA-hTRIM28(S473A).

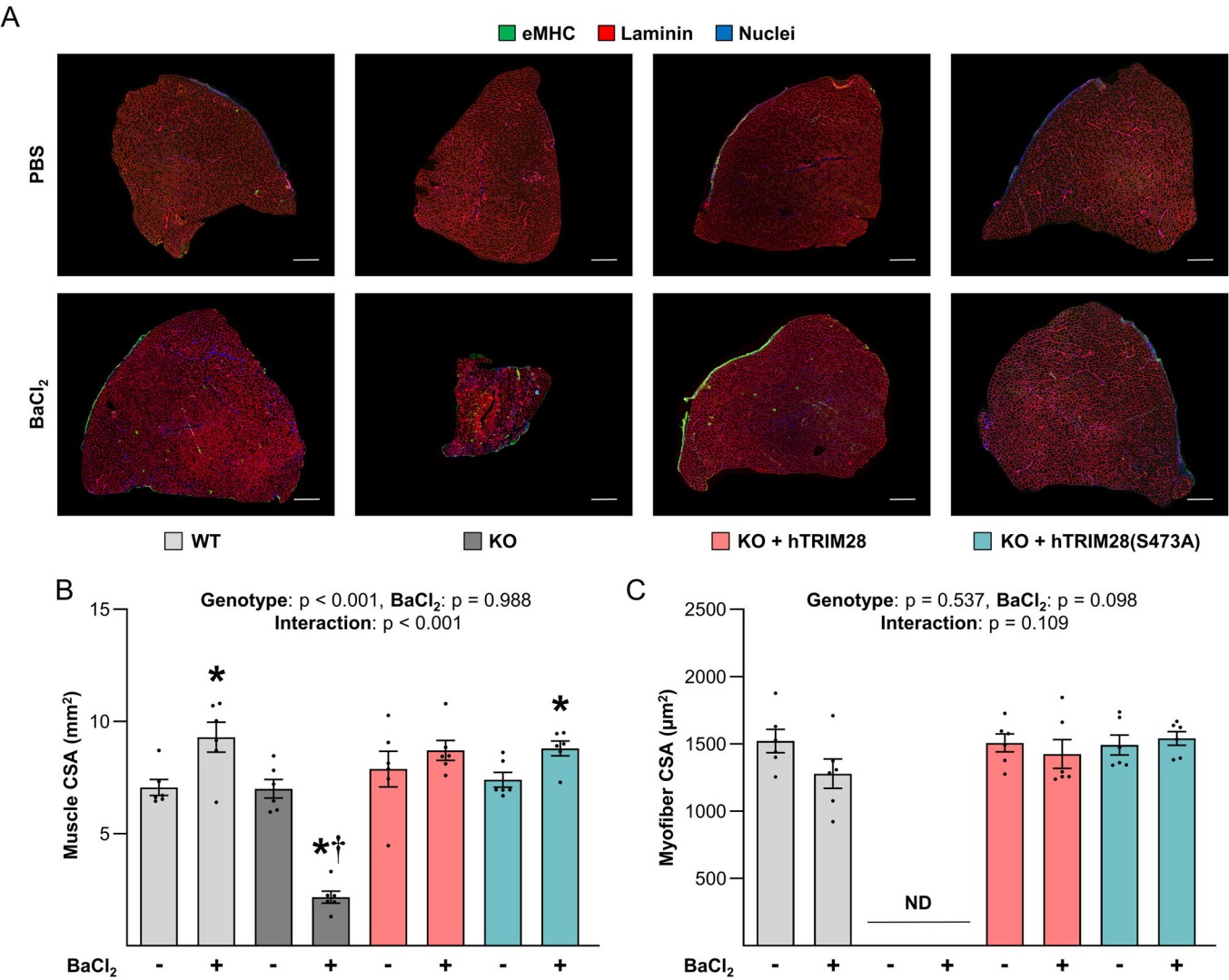

**Figure EV5. TRIM28(S473) phosphorylation in satellite cells is not required for the restoration of muscle CSA or myofiber CSA following BaCl$_2$-induced injury.**

At 14 days post tamoxifen the tibialis anterior muscles of wild-type (WT) mice, tamoxifen-inducible and satellite cell-specific TRIM28 knockout mice (KO) mice, as well as KO mice that contain tamoxifen-inducible "rescue" expression of hTRIM28 (KO + hTRIM28) or phosphodefective hTRIM28 (KO + hTRIM28(S473A)) were injected with BaCl$_2$ (+) to induce injury or PBS (−) as a control condition. The tibialis anterior muscles were collected after a 21-day recovery period. (**A**) Mid-belly cross sections of the muscles were subjected to immunohistochemistry for eMHC, laminin, and nuclei. Scale bar = 500 μm. (**B**) Measurements of whole muscle cross-sectional area (CSA) (two-way repeated-measures ANOVA, $n = 6$/group, *$p < 0.001$, = 0.002, = 0.036, †$p < 0.001$), and (**C**) the mean myofiber CSA per muscle (two-way repeated-measures ANOVA, $n = 6$/group). Values are presented as the group mean ± SEM, $n = 6$/group. Due to the absence of clear myofibers in the BaCl$_2$-treated muscles of KO mice, myofiber CSA data for these mice was not determined (ND). * indicates a significant effect of BaCl$_2$ within a given genotype, † indicates significant difference from the BaCl$_2$ treated WT condition, $p < 0.05$.

