## [Peer Review File · EMBO Reports]

Satellite cell-derived TRIM28 is pivotal for mechanical load- and injury-induced myogenesis

Kuan-Hung Lin, Jamie Hibbert, Corey Flynn, Jake Lemens, Melissa Torbey, Nathaniel Steinert, Philip Flejsierowicz, Kiley Melka, Garrison Lindley, Marcos Lares, Vijayasradhi Setaluri, Amy Wagers, and Troy Hornberger

Corresponding author(s): Troy Hornberger (troy.hornberger@wisc.edu)

Review Timeline:

Submission Date:	4th Jan 24
Editorial Decision:	27th Feb 24
Revision Received:	31st May 24
Editorial Decision:	3rd Jul 24
Revision Received:	19th Jul 24
Accepted:	26th Jul 24

Transaction Report:

Dear Dr. Hornberger

Thank you for the submission of your research manuscript to our journal. I am sorry for the delay in handling your manuscript, but we have only now received the final referee report. All reports are copied below.

As you will see, the referees acknowledge that the findings are potentially interesting, but they also raise a number of concerns and have suggestions how to further strengthen the conclusions that are all pertinent and need to be addressed. Please also reconcile your findings with earlier data that TRIM28 regulates metabolism, proteostasis and autophagy in muscle fibres and replicate the C2C12 work.

Given these constructive comments, we would like to invite you to revise your manuscript with the understanding that the referee concerns (as detailed above and in their reports) must be fully addressed and their suggestions taken on board. Please address all referee concerns in a complete point-by-point response. Acceptance of the manuscript will depend on a positive outcome of a second round of review. It is EMBO Reports policy to allow a single round of revision only and acceptance or rejection of the manuscript will therefore depend on the completeness of your responses included in the next, final version of the manuscript.

We realize that it is difficult to revise to a specific deadline. In the interest of protecting the conceptual advance provided by the work, we recommend a revision within 3 months (May 27). Please discuss the revision progress ahead of this time with the editor if you require more time to complete the revisions.

I am also happy to discuss the revision further via e-mail or a video call, if you wish.

*****IMPORTANT NOTE:

We perform an initial quality control of all revised manuscripts before re-review. Your manuscript will FAIL this control and the handling will be delayed IN CASE the following APPLIES:

- 1) A data availability section providing access to data deposited in public databases is missing. If you have not deposited any data, please add a sentence to the data availability section that explains that.
- 2) Your manuscript contains statistics and error bars based on $n=2$. Please use scatter blots in these cases. No statistics should be calculated if $n=2$.

When submitting your revised manuscript, please carefully review the instructions that follow below. Failure to include requested items will delay the evaluation of your revision.*****

- 2) individual production quality figure files as .eps, .tif, .jpg (one file per figure).

Please download our Figure Preparation Guidelines (figure preparation pdf) from our Author Guidelines pages <https://www.embopress.org/page/journal/14693178/authorguide> for more info on how to prepare your figures.

- 4) a complete author checklist, which you can download from our author guidelines

(<<https://www.embopress.org/page/journal/14693178/authorguide>>). Please insert information in the checklist that is also reflected in the manuscript. The completed author checklist will also be part of the RPF.

5) Please note that all corresponding authors are required to supply an ORCID ID for their name upon submission of a revised manuscript (<<https://orcid.org/>>). Please find instructions on how to link your ORCID ID to your account in our manuscript tracking system in our Author guidelines (<<https://www.embopress.org/page/journal/14693178/authorguide#authorshipguidelines>>)

6) We replaced Supplementary Information with Expanded View (EV) Figures and Tables that are collapsible/expandable online. A maximum of 5 EV Figures can be typeset. EV Figures should be cited as 'Figure EV1, Figure EV2' etc... in the text and their respective legends should be included in the main text after the legends of regular figures.

7) Please note that a Data Availability section at the end of Materials and Methods is now mandatory. In case you have no data that requires deposition in a public database, please state so instead of refereeing to the database. See also < <https://www.embopress.org/page/journal/14693178/authorguide#dataavailability>>. Please note that the Data Availability Section is restricted to new primary data that are part of this study.

Additional information on source data and instruction on how to label the files are available <<https://www.embopress.org/page/journal/14693178/authorguide#sourcedata>>.

10) Figure legends and data quantification:
The following points must be specified in each figure legend:

- the name of the statistical test used to generate error bars and P values,
 - the number (n) of independent experiments (please specify technical or biological replicates) underlying each data point,
 - the nature of the bars and error bars (s.d., s.e.m.)
- If the data are obtained from n {less than or equal to} 5, show the individual data points in addition to the SD or SEM.
- If the data are obtained from n {less than or equal to} 2, use scatter blots showing the individual data points.

See also the guidelines for figure legend preparation:
<https://www.embopress.org/page/journal/14693178/authorguide#figureformat>

11) Our journal encourages inclusion of *data citations in the reference list* to directly cite datasets that were re-used and obtained from public databases. Data citations in the article text are distinct from normal bibliographical citations and should directly link to the database records from which the data can be accessed. In the main text, data citations are formatted as follows: "Data ref: Smith et al, 2001" or "Data ref: NCBI Sequence Read Archive PRJNA342805, 2017". In the Reference list, data citations must be labeled with "[DATASET]". A data reference must provide the database name, accession number/identifiers and a resolvable link to the landing page from which the data can be accessed at the end of the reference. Further instructions are available at <<https://www.embopress.org/page/journal/14693178/authorguide#referencesformat>>.

12) All Materials and Methods need to be described in the main text. We would encourage you to use 'Structured Methods', our new Materials and Methods format. According to this format, the Materials and Methods section should include a Reagents and Tools Table (listing key reagents, experimental models, software and relevant equipment and including their sources and relevant identifiers) followed by a Methods and Protocols section in which we encourage the authors to describe their methods

using a step-by-step protocol format with bullet points, to facilitate the adoption of the methodologies across labs. More information on how to adhere to this format as well as downloadable templates (.doc or .xls) for the Reagents and Tools Table can be found in our author guidelines: <

<https://www.embopress.org/page/journal/14693178/authorguide#manuscriptpreparation>>. An example of a Method paper with Structured Methods can be found here: <<https://www.embopress.org/doi/10.15252/msb.20178071>>.

13) As part of the EMBO publication's Transparent Editorial Process, EMBO Reports publishes online a Review Process File to accompany accepted manuscripts. This File will be published in conjunction with your paper and will include the referee reports, your point-by-point response and all pertinent correspondence relating to the manuscript.

Yours sincerely,

Referee #1:

Background:

Understanding the cellular and molecular mechanisms that regulate muscle growth is an important topic.

In the current manuscript, the authors propose a mechanistic link between TRIM28 and myomixer.

Previous work from King et al (2023) used a developmental and adult specific muscle deletion model and observed that TRIM28 regulated metabolism and autophagy in muscle fibers. In a FASEB abstract by Seinert et al., 2021. The Hornberger group described that TRIM28 regulated METT21 members to regulate protein degradation machinery. This highlights an important role of TRIM28 in adult muscle fibers.

In this manuscript the authors examine the role of TRIM28 in vivo. This is the first description of the important role of TRIM28 in murine muscle stem cells and its interaction with Myomixer. Therefore, this finding is novel.

Previous work has shown a role of TRIM28 phosphorylation on myogenic differentiation in an immortalized cell line. Therefore the role of TRIM28 role in myogenesis is not novel, however the authors argue that this is dispensable in primary muscle cells in vitro and in vivo. This discrepancy, if true is an important finding about the context dependence of TRIM28 and the potential incorrect conclusions one makes when using immortalized cell lines.

In conclusion the main point of novelty for this manuscript is the role of TRIM28 independent of serine 473 phosphorylation. Additional controls and in vitro experiments are required to bolster the claims around TRIM28 phosphorylation. The link to myomixer could be further developed, and the relationship to metabolism and autophagy needs to be considered.

Results in the present manuscript:

In the present manuscript, Lin et al., test for the first time the role of TRIM28 in satellite cell mediated muscle growth and repair in vivo.

The authors reveal that during earlier stages of repair, TRIM28 is dispensable for SC proliferation and early differentiation, whereas there an extensive defect in the latter stages of muscle repair, at the stage of muscle fusion.

Using in vitro myotube assays the authors showing a robust defect in myotube fusion consistent with the in vivo results.

Previous work from the Dilworth lab demonstrated that TRIM28 was required for myoblast differentiation via Myod using immortalized C2C12 cells. Moreover, TRIM28 impact on myod-dependent differentiation in C2C12 cells was via Phosphorylation of TRIM28 at serine 473.

This is inconsistent with the results in the present manuscript. Due to technical reasons the authors were not able to repeat the same experiment.

In the present manuscript the authors asked whether phosphorylation of serine 473 was required for fusion defects using primary myoblasts and using a novel in vivo transgenic rescue approach.

The results demonstrate that add-back of human TRIM28 or hTRIM28 non-phospho form was able to rescue the primary defect in absence of mouse TRIM28. Based on the results, the authors correctly conclude that is not required for TRIM28 role in regenerative myogenesis.

Finally, the authors investigate potential downstream effectors of TRIM28, with a focused attention on Myomaker and Myomixer. Based on correlated protein expression between TRIM28 and Myomixer, phenotype of pore formation the authors conclude that TRIM28 is regulating Myomixer expression to control fusion.

Previous work from King et al (2023) used a developmental and adult specific muscle deletion model and observed that TRIM28 regulated metabolism and autophagy in muscle fibers. In a FASEB abstract by Seinert et al., 2021. The Hornberger group described that TRIM28 regulated METT21 members to regulate protein degradation machinery. Therefore without further mechanistic insight into how TRIM28 is regulating myomixer its plausible that TRIM28 is regulating previously described processes to control myotube fusion. Both papers should be cited and discussed.

There are four areas that require attention:

- 1) It is essential that the authors replicate the C2C12 work. In its absence it remains unknown why these results are discrepant. Having these two discrepant findings in the literature does more harm than good for scientific consensus.
- 2) To be confident of the transgenic rescue please show TRIM28 IHC staining in control, TRIM28 cKO and hTRIM28 rescue. Is there a way to confirm human TRIM28 expression or the serine 473 phospho activity? If so, please provide that.
- 3) Additional experiments are required to flush out the mechanism of TRIM28 regulation of myomixer.
 - a. Show protein expression between TRIM28 and myomaker.
 - b. Show myomaker expression after BaCl injury.
 - c. Show transcript levels of myomixer and myomaker in control and nulls after injury.
 - d. Overexpress TRIM28 in vitro to show myomaker and myomaker transcript and protein expression. If this cannot be performed perform the measures after hTRIM28 rescue.
- 4) The authors need to address the prior literature from their own group and King et al., to integrate the findings that TRIM28 could regulate fusion and muscle fiber defects during repair via METTL21, metabolism or autophagy.

Referee #2:

In this study, Lin and colleagues investigated the roles of TRIM28 in two myogenesis models, muscle regeneration and hypertrophy. In contrast with the published data showing the critical roles of phosphorylation in TRIM28 S473 and its roles in the suppression of myogenin, here, the authors demonstrated the critical roles of TRIM28 in cell fusion via independent of the phosphorylation in S473. Further, the authors presented data indicating that the loss of TRIM28 resulted in a decrease in the expression of Myomixer, but not Myomaker. Overall, the data are convincing, and the manuscript is well-written. The authors provide strong evidence supporting the phosphorylation-independent roles of TRIM28 in myoblast-myotube fusion. This study will provide a new insight into the research field.

I suggest additional experiments to further strengthen the findings.

- 1) Figure 2; I The short duration (2 weeks) may hinder accurate observation of muscle weight and CSA differences between control and low myonuclear accretion mice due to edema, disrupting true hypertrophic responses. Generally, 8 weeks are

necessary for obtaining conclusive data. Because the relevance between the unusual staining of dystrophin and TRIM28 function is unclear, the reviewer suggests including SA-8 weeks experiment results. If additional experiments are not feasible promptly (e.g., mouse preparation issues), cite the following paper or other sources and describe why there was no difference in CSA or muscle weights between WT and KO.

F. Damas, C.A. Libardi, C. Ugrinowitsch, The development of skeletal muscle hypertrophy through resistance training: the role of muscle damage and muscle protein synthesis, *Eur. J. Appl. Physiol.* 118 (3) (2018) 485-500.

Fry CS, Lee JD, Jackson JR, Kirby TJ, Stasko SA, Liu H, Dupont-Versteegden EE, McCarthy JJ, Peterson CA. Regulation of the muscle fiber microenvironment by activated satellite cells during hypertrophy. *The FASEB Journal* 28 (2014) 1654-1665.

2) Figure 4; As the authors described, the eMyHC+/regenerating myotube seems to disappear between days 5 and 21 post-njury in KO mice. Are there myoblasts undergoing apoptosis without fusion? In addition, does the eMyHC+ myotube also die without TRIM28 or additional fusion?

3) Figures 7 and 8 (in vivo rescue experiments); As shown in Figure 1, the authors need to confirm the presence or absence of P-TRIM28 using the specific antibody.

4) Figure 10E; Is the expression of MYOD and MYOGENIN in KO comparable to that in WT mice? Such data will strengthen that TRIM28 does not impact the expression of MYOD and MYOGENIN in vivo.

Referee #3:

Overall this is a very well written manuscript describing for the first time the importance of TRIM28 in the regulation of a specific step in myogenic fusion through the regulation of MYMX. Although the exact mechanism through which TRIM28 does this action remains unclear, the authors demonstrated through ample evidence that TRIM28 is a critical component. The authors is cognizant of the work that remains to be done to elucidate the mechanism as a limitation of their study. As such, I found this study highly suitable for publication. The majority of the manuscript and figures was clear with some minor issues identified below.

Remarks to the Author.

Please consider generating a new Figure 1A, which seems to be a reproduction of Figure 1 in Baar and Esser, 1999.

Results section states "As shown in Fig. 1F-G, 114 immunohistochemical analysis revealed that >98% of the satellite cells in the plantaris muscles from the KO mice were TRIM28 negative, whereas the expression of TRIM28 remained readily detectable in Pax7 negative cells." (lines 114-117) but the Figure 1G shows >98% satellite cells in the WT is TRIM28+ and no quantification of the number of pax7 negative nuclei. Considering revising this statement and/or figure/legend accordingly.

Strongly encourage including individual values for bar graphs.

Figure S3A lacks dystrophin which is shown in Figures 3A and S4A. Is there a particular reason for this? Seems the quantification was performed without taking fibers into consideration.

Figure S9B needs markers for band sizes.

The in vivo rescue of TRIM28 with and without S473 was well done.

Authors' responses

Reviewer comments are in black italics; our responses are in plain blue text.

Referee #1:

1) It is essential that the authors replicate the C2C12 work. In its absence it remains unknown why these results are discrepant. Having these two discrepant findings in the literature does more harm than good for scientific consensus.

Serial passage of immortalized cell lines, including C2C12 myoblasts, is known to cause differences in phenotype / genotype. Thus, the same type of cell line could greatly differ amongst different labs. On this note, the C2C12 myoblasts we used in our first attempts were from Dr. Hornberger's lab, and the experiments were performed at the University of Wisconsin – Madison. To address the reviewer's request and the point mentioned above, we tried to perform the C2C12-based studies again with C2C12 myoblasts from Dr. Wagers's lab, and this time the experiments were performed at Harvard University. Consistent with our original results, we were able to successfully select lentivirus-transduced C2C12 cells from control (CNT) and TRIM28 knockdown (KD) lentivirus-infected cells, but transduction rates were extremely low in cells that were infected with either the TRIM28 knockdown + wild-type "rescue" (KD + WT) or the TRIM28 knockdown + phosphodeficient "rescue" (KD + PD) lentivirus. This point is illustrated below in which images of the infected cells were captured 3 days after puromycin selection, scale bar = 50 μ m.

We appreciate why the reviewer would like for us to be able to replicate the C2C12-based studies that were performed by Singh et al., and we would like to reiterate that the authors of that paper provided us with the constructs that they used in their original study. However, we have now attempted to repeat their studies in two different labs and although we were able to successfully use the constructs in primary myoblasts (see Appendix Figure S6), the constructs are simply not working for us in C2C12s. We hope that the reviewer can appreciate that we have earnestly attempted to replicate the C2C12 studies, but in our view, there is nothing further that we can do to address the request.

2) To be confident of the transgenic rescue please show TRIM28 IHC staining in control, TRIM28 cKO and hTRIM28 rescue. Is there a way to confirm human TRIM28 expression or the serine 473 phospho activity? If so, please provide that.

In Figure 8C, the detection of HA-positive nuclei confirms that there was expression of HA-tagged human TRIM28 in both the wild-type and phosphodeficient rescue conditions. We recognize that it would be ideal to have images that confirm a lack of serine 473 phosphorylation in the nuclei that express the phosphodeficient

variant. Indeed, when our lab started this project, there was a polyclonal antibody available for detecting TRIM28(S473) phosphorylation. The antibody worked extremely well and it was used to generate the images in Figure 1. Unfortunately, however, the serum that was used to generate that antibody ran out a few years ago. The company that was distributing the antibody tried to generate a replacement, but their attempts to create a suitable replacement failed and the antibody was discontinued. We have tested all other commercially available antibodies, and unfortunately, none of them have proven to be capable of detecting TRIM28(S473) phosphorylation when appropriate positive and negative controls are employed. We have also spoken with several companies about generating a custom antibody. All the companies we spoke with indicated that, given the sequence surrounding the phosphosite, the chance of generating an antibody with a high degree of specificity was quite low. This is important because the generation of a custom antibody can be very expensive (apx. \$25,000 USD for a monoclonal) and we simply cannot risk that kind of expenditure when we know that the chances of success are low. Therefore, given our inability to confirm the phosphodeficient mutation with an antibody, we deferred to the use of a genetic approach. This approach is shown in Figure EV4 and provides genetic confirmation that the phosphodeficient mutant possessed the appropriate modification in the coding sequence.

3) Additional experiments are required to flush out the mechanism of TRIM28 regulation of myomixer.

a. Show protein expression between TRIM28 and myomaker.

We thank the reviewer for the suggestion and have added the scatter plot in Figure 10D of the revised manuscript. Unlike MYMX, we did not see a significant correlation between the levels of MYMK and TRIM28 during myotube formation. To clarify this, we added the following statement (lines 469- 471), - “We also discovered that the differentiation-induced increase in MYMX was very highly correlated with the level of TRIM28 in the cells (Figure 10C and E), but no significant correlation was detected between the levels of TRIM28 and MYMK in the cells (Figure 10B and D).”

b. Show myomaker expression after BaCl injury.

This is a great suggestion. We checked the MYMK expression level after muscle injury and found that, unlike MYMX, the loss of TRIM28 in satellite cells does not affect the MYMK expression level at 3 days post-injury. This result is now included in the revised Figure 10F and I.

c. Show transcript levels of myomixer and myomaker in control and nulls after injury.

Although we showed that TRIM28 controls the level of MYMX during myogenesis in this study, it is not clear whether it is through transcriptional regulation. It is well known that TRIM28 is a transcription intermediate factor, but it has also been shown to play a role in controlling protein degradation. Additionally, studying whether TRIM28 is required for the increase in *Mymx* and *Mymk* transcript levels in satellite cells/myoblasts upon muscle injury at the whole muscle level could be misleading because the complex cell composition and transcript abundance in different cell types may not be comparable between injured WT and KO muscles. As a supporting example, a previous study showed that the loss of MYMX does not impair the increase in *Mymk* transcript level in myoblasts during myotube formation (PMID 28569755 - note: Myomixer is an alias of MYMX). However, the loss of MYMX significantly attenuated the increase in the level of *Mymk* transcripts in injured muscles that were

undergoing regeneration (PMID 29581287). For the reviewer's convenience, the results from the two studies are shown below. Our results show that the loss of TRIM28 leads to an impaired increase in MYMX, and given the results in PMID 29581287, it would not be surprising if we also detected an attenuated injury-induced increase in *Mymk* expression in our KO mice but, as illustrated above, this would not necessarily mean that TRIM28 is required for injury-induced increase in *Mymk* expression in satellite cells/myoblasts. As such, we do not believe that performing the suggested experiment would further provide mechanistic insight into how TRIM28 regulates myogenesis. In our opinion, in order to properly determine whether TRIM28 is required for the induction of *Mymk* and *Mymx* transcripts during muscle regeneration, the comparison between injured WT and KO muscles would have to be made at a single cell/nucleus scale specifically and specifically in the satellite cell/myoblast cell population. Such an approach is beyond the scope of the current study.

Figure for referees is redacted.

d. Overexpress TRIM28 in vitro to show myomaker and myomaker transcript and protein expression. If this cannot be performed perform the measures after hTRIM28 rescue.

We do not understand why the reviewer would like us to perform this experiment. Specifically, during myotube formation (Figure 6A-B), we actually detected a decrease in the protein level of TRIM28 at 2 days post-differentiation (a time point at which MYMK and MYMX expression were significantly increased). This result suggests that TRIM28 does not regulate myotube formation through a mechanism that involves an increase of its own protein level. Thus, the rationale for wanting us to perform overexpression studies is not clear.

4) The authors need to address the prior literature from their own group and King et al., to integrate the findings that TRIM28 could regulate fusion and muscle fiber defects during repair via METTL21, metabolism or autophagy.

We thank the reviewer for highlighting the opportunity to integrate other functions of TRIM28 into this manuscript. However, the previous work by our lab and King et al. were done in myofiber-specific TRIM28 knockout mice. As such, we are not convinced that it would be appropriate to draw comparisons between the different studies. Additionally, in King et al.'s study, it does not seem like the myofiber-specific loss of TRIM28 impacts metabolism. For the reviewer's convenience, we quote the statements from the abstract of King et al.'s

study “We studied two different muscle-specific (MCK-Cre and ACTA1-Cre-ERT2) TRIM28 knockout models, which were phenotyped during and after being fed a chow or high-fat diet (HFD). Whilst muscle-specific deletion of TRIM28 in both models demonstrated alterations in markers of mitochondrial activity and autophagy in skeletal muscle, we did not observe major impacts on the majority of metabolic measures in these mice.”

In an effort to fulfill the reviewers request we tried to find places in the manuscript in which we could discuss the previous findings about TRIM28 and its relation to METTL21, protein degradation, and autophagy. However, we could not find a logical place for the discussion of these topics. Indeed, the only place we found that would not severely disrupt the flow of the manuscript was in the limitations section. Specifically, in the limitations section we highlight the fact that we do not know the mechanisms via which TRIM28 regulates the expression of MYMX and concomitant fusion pore formation. In our original version of this section, we intentionally took a very broad approach when discussing possible mechanisms. For instance, we considered the possibility of transcriptional regulation as well as the different types of post-translational modifications that TRIM28 could confer (e.g. SUMO E3 ligase and ubiquitin E3 ligase activities). We have decided to stick with the original approach because any kind of discussion about how TRIM28 might use METTL21, autophagy, or protein degradation to regulate MYMX and fusion pore formation would seem forced, and it would be based purely on speculation. Nonetheless, if the reviewer can point us towards a specific location in which he/she feels that a discussion about the topics of METTL21, protein degradation, and/or autophagy is warranted, then we would be more than happy to reconsider our position.

Referee #2:

1) Figure 2; The short duration (2 weeks) may hinder accurate observation of muscle weight and CSA differences between control and low myonuclear accretion mice due to edema, disrupting true hypertrophic responses. Generally, 8 weeks are necessary for obtaining conclusive data. Because the relevance between the unusual staining of dystrophin and TRIM28 function is unclear, the reviewer suggests including SA-8 weeks experiment results. If additional experiments are not feasible promptly (e.g., mouse preparation issues), cite the following paper or other sources and describe why there was no difference in CSA or muscle weights between WT and KO.

We appreciate that models of chronic mechanical loading can lead to edema and that such an effect could obscure true hypertrophic responses (particularly in the first few days after the onset of increased loading). However, the 2 week time point has been used in many of the notable studies that examined the interplay between chronic mechanical loading, impaired myonuclear accretion, and the induction of hypertrophy (PMID:28186492, 21828094, 27531949, and 22225874). Our lab has also used the 2 week time point in numerous studies that were aimed at defining the mechanism(s) via which an increase in mechanical loading induces hypertrophy. Importantly, we have not obtained evidence to support the notion that the hypertrophy we observe is due to edema. For instance, in the current study we observed hypertrophy of the type IIa and type IIx fibers but not the type IIb fibers (Fig. EV1). If the hypertrophy was due to edema, then we would have expected to see this effect in all fiber types, but this was not the case. In a previous study we also used our model to

demonstrate that mTOR, within the skeletal muscle fibers themselves, is the rapamycin-sensitive element that confers the hypertrophic response (PMID: 21946849). If the hypertrophy that we observed in that study was due to edema, then it would mean that the edema was driven by mTOR within the skeletal muscle fibers themselves. While this is certainly possible, we are not aware of any studies that would support that notion. Finally, we recently developed a method that can be used to visualize myofibrils (FIM-ID) and we used this method to demonstrate that the increase in myofiber CSA that occurs after 16 days of mechanical loading is largely mediated by an increase in the number of myofibrils per muscle fiber (i.e., myofibrillogenesis) (PMID: 38466320). We cannot imagine a situation in which edema would lead to the appearance of more myofibrils and therefore it is our conviction that the mechanical load-induced growth that we observed in the KO mice was a true hypertrophic response.

2) *Figure 4; As the authors described, the eMyHC+/regenerating myotube seems to disappear between days 5 and 21 post-njury in KO mice. Are there myoblasts undergoing apoptosis without fusion? In addition, does the eMyHC+ myotube also die without TRIM28 or additional fusion?*

These are excellent questions and to address them we performed a TUNEL assay on injured WT and KO muscles at 10 days post-injury. The figure below has been added as Appendix Fig. S5, and as shown in this figure, there were significantly more TUNEL-positive nuclei in the KO muscles than the WT muscles at 10 days post-injury. Further, some of these TUNEL-positive nuclei were found inside of the laminin positive / multinucleated nascent myofibers. Thus, the results of this assay provide support for the notion that at least some of the eMHC+ myoblasts/myofibers in KO muscles underwent cell death.

3) *Figures 7 and 8 (in vivo rescue experiments); As shown in Figure 1, the authors need to confirm the presence or absence of P-TRIM28 using the specific antibody.*

We fully understand the basis for this request and as shown in Figure 8C, the detection of HA positive nuclei confirms that there was expression of HA-tagged human TRIM28 in both the wild-type and phosphodeficient rescue conditions. We recognize that it would be ideal to have images that confirm a lack of serine 473 phosphorylation in the nuclei that express the phosphodeficient variant. Indeed, when our lab started this project, there was a polyclonal antibody available for detecting TRIM28(S473) phosphorylation. The antibody worked extremely well and it was used to generate the images in Figure 1. Unfortunately, the serum

that was used to generate that antibody ran out a few years ago. The company that was distributing the antibody tried to generate a replacement, but their attempts to create a suitable replacement failed and the antibody was discontinued. We have tested all other commercially available antibodies, and unfortunately, none of them have proven to be capable of detecting TRIM28(S473) phosphorylation when appropriate positive and negative controls are employed. We have also spoken with several companies about generating a custom antibody. All the companies we spoke with indicated that, given the sequence surrounding the phosphosite, the chance of generating an antibody with a high degree of specificity was quite low. This is important because the generation of a custom antibody can be very expensive (apx. \$25,000 USD for a monoclonal) and we simply cannot risk that kind of expenditure when we know that the chances of success are low. Therefore, given our inability to confirm the phosphodeficient mutation with an antibody we deferred to a genetic approach. This approach is shown in Figure EV4 and provides genetic confirmation that the phosphodeficient mutant possessed the appropriate modification in the coding sequence.

4) *Figure 10E; Is the expression of MYOD and MYOGENIN in KO comparable to that in WT mice? Such data will strengthen that TRIM28 does not impact the expression of MYOD and MYOGENIN in vivo.*

This is a great question. We checked the MYOD and MYOG levels and found that the induction of MYOG, but not MYOD, is moderately attenuated in KO mice (Figure 10F-H). This is very similar to what we observed when primary myoblasts were cultured in differentiation media for 2 days (Figure 6). However, our *in vivo* results also showed that the number of MYOG positive cells/nuclei is not lower in KO muscles after mechanical load and is dramatically higher at 10 days post-injury (Figure 2C-D and Appendix Fig. S3A-B). Together, these results suggest that the attenuated induction of MYOG is caused by lower MYOG expression in each cell. MYOG is known to boost myoblast fusion through a MYMK-dependent mechanism. Nevertheless, the unaffected induction of MYMK in KD primary myoblasts during myotube formation and KO muscles during regeneration suggests that the attenuated MYOG level was still sufficient to induce MYMK expression. These results, along with our observation that the KD primary myoblast can still undergo hemifusion, indicate that TRIM28 is not necessary for a MYOG-MYMK dependent mechanism in myoblast fusion. Thus, even though the loss of TRIM28 leads to attenuated MYOG expression during myogenesis, our main conclusion that “TRIM28 controls the expression of MYMX and concomitant fusion pore formation” is still accurate. To address the new data showing the attenuated induction of MYOG in KO injured muscles, we added the following statements in the manuscript:

(Lines 308309) - “However, the knockdown of TRIM28 did lead to a trend towards a decrease in differentiation-induced expression of MYOG (Fig. 6D).”

(Lines 472-480) - “To determine whether similar observations would be made *in vivo*, we examined the level of MYOD, MYOG, MYMK, and MYMX in the TA muscles of WT and KO mice that had been injured with BaCl₂. As shown in Fig. 10F-J, the outcomes were very similar to what was observed in the primary myoblasts with injury leading to an increase in all of the examined proteins. Specifically, the loss of TRIM28 did not affect the injury-induced increases in MYOD and MYMK, whereas the increase in MYOG was partially attenuated and the increase in MYMX was largely abolished. Thus, the results of our analyses, along with the *a priori* knowledge about MYMX, have led us to the conclusion that TRIM28 regulates myogenesis by controlling the induction of MYMX expression and concomitant fusion pore formation.”

Referee #3:

Please consider generating a new Figure 1A, which seems to be a reproduction of Figure 1 in Baar and Esser, 1999.

We agree with the reviewer's concern and removed panel A from the original version of Figure 1.

Results section states "As shown in Fig. 1F-G, 114 immunohistochemical analysis revealed that >98% of the satellite cells in the plantaris muscles from the KO mice were TRIM28 negative, whereas the expression of TRIM28 remained readily detectable in Pax7 negative cells." (lines 114-117) but the Figure 1G shows >98% satellite cells in the WT is TRIM28+ and no quantification of the number of pax7 negative nuclei. Considering revising this statement and/or figure/legend accordingly.

We thank the reviewer for pointing out that we did not include data to support the statement for Pax7 negative nuclei. To address this, we quantified the % of Pax7 negative nuclei that are positive for TRIM28 and provided the result in revised Figure 1G.

Strongly encourage including individual values for bar graphs.

As suggested, all bar graphs now include the individual values.

Figure S3A lacks dystrophin which is shown in Figures 3A and S4A. Is there a particular reason for this? Seems the quantification was performed without taking fibers into consideration.

In this figure (formerly S3 now titled Appendix Fig. S2), the muscles were harvested at 3 days post-injury, and we did not yet see obvious myofiber formation. This is consistent with previous studies showing the kinetics of muscle regeneration (PMID 14627628, 26807982, and 35898401). Because of this, mid-belly cross sections were not co-stained with dystrophin.

Figure S9B needs markers for band sizes.

We thank the reviewer for the suggestion and have now amended the image to show the DNA ladder in the revised figure that is now Figure EV4.

The in vivo rescue of TRIM28 with and without S473 was well done.

We appreciate the reviewer's kind appraisal.

Dear Dr. Hornberger

Thank you for the submission of your revised manuscript to EMBO reports. Referee #1 was unfortunately not available to review the revised version and I have therefore asked referee #2 to assess your response to the concerns raised by referee #1 as well.

Referee #2 kindly agreed to do so and considered your response to the concerns from referee #1 overall adequate. Regarding Ref#1, Comment #1 the referee acknowledged that you have obtained C2C12 cells from another laboratory and that you show similar experiments using primary myoblasts, so the study should not be rejected because of a discrepancy with the previous report.

Referee #2 however also noted that Comments #2 and #3 from referee #1 were not addressed because of the lack of a specific antibody for the phosphorylated TRIM28 variant. Referee #2 him/herself remains concerned about the lack of evidence that the point mutation was successfully generated (see below). These remaining concerns must be addressed as outlined by Referee #2.

From the editorial side, there are also a few things that we need before we can proceed with the official acceptance of your study.

- Please note that we have accepted the tracked changes and uploaded a clean manuscript file; the track-changes one was uploaded as Related Manuscript file in the online manuscript tracking system.
- Please provide up to 5 keywords.
- 'Materials and Methods' should be 'Methods'.
- The Appendix section (line 1385-87) should be removed from the manuscript.
- The manuscript sections should be in the following order: Title page - Abstract & Keywords - Introduction - Results - Discussion - Methods - Data Availability - Acknowledgments - Disclosure Statement & Competing Interests - References - Figure Legends - (Main Tables with legends) - Expanded View Figure Legends.
- Please update the 'Conflict of interest' paragraph to our new 'Disclosure and competing interests statement'. For more information see <https://www.embopress.org/page/journal/14693178/authorguide#conflictsofinterest>
- Regarding the Author Contributions, we now use CRediT to specify the contributions of each author in the journal submission system. Therefore, please remove the Author Contributions from the manuscript file and make sure that the author contributions in our online manuscript tracking system are correct and up-to-date. The information you specified in the system will be automatically retrieved and typeset into the article. You can enter additional information in the free text box provided, if you wish.
- The funding information needs to be part of the Acknowledgments
- Please update the references to the alphabetical Harvard style. The abbreviation 'et al' should be used if there are more than 10 authors. You can download the respective EndNote file from our Guide to Authors https://endnote.com/style_download/embo-reports/
- Please note that all data mentioned must be included in the manuscript. In this regard, we note 'data not shown' on page 24, legend for Figure 3F. Please either include the relevant data, e.g., in the Appendix, or remove the statement on "No difference between the conditions..."
- Callouts in the text are missing for Fig. 3DEF, Fig. 5ABCD, and Fig. 7AB. Please insert callouts to these panels in the text where appropriate. The callout "Table S1" needs to be corrected. I assume this refers to Table EV1, which lists the antibodies?
- That said, we encourage authors (and require it from July 1st onwards) to use our Reagents and Tools table (listing key reagents, experimental models, software and relevant equipment and including their sources and relevant identifiers). The aim is to facilitate adoption of the methodologies across labs. More information on how to adhere to this format as well as a downloadable template (.docx) for the Reagents and Tools Table can be found in our author guidelines: <https://www.embopress.org/page/journal/14693178/authorguide#structuredmethods>.

An example of a Method paper with Structured Methods can be found here: <https://www.embopress.org/doi/10.15252/msb.20178071>.

- Appendix: please add page numbers for each item/figure in the table of contents.
 - Data availability section: please remove "and materials" from the heading and please provide a link that directly resolves on the dataset, S-BIAD1194, on BioImage Archive.
 - Our production/data editors have asked you to clarify several points in the figure legends (see below). Please incorporate these changes in the manuscript and return the revised file with tracked changes with your final manuscript submission.
 - A) Please note that the exact p values are not provided in the legends of figures 1b, d, f; 2b, d; 2b-c, e-f; 4f; 5b-d; 6b, d, g-h; 7e; 8b, d; 9d, g; 10c, e, g-h, j; EV 1b-c; EV 2b, d, f; EV 3b-f; EV 5b.
 - B) Please indicate the statistical test used for data analysis in the legends of figures 10d-e.
 - I have modified the synopsis summary text, mainly to shorten it. Can you please review the changes in the attached file? Thank you.
 - We perform a routine image analysis on all manuscripts prior to publication. In this context we noticed the following aberrations in Figure 4:
 - A portion of the WT section at 10 dpi from Fig. 4A is shown again in Fig. 4C. The same holds true for 4A, KO, 7 dpi and Fig. 4D. And the image from Fig. 4D is shown again in Fig. 4E.
 - I assume these are zooms on a certain type of myoblast/myofiber observed in these sections. If so, please state this in the figure legend and indicate the zoomed areas with boxes in (A).
 - The source data for Fig. 6A, MyoD blot, seems not to match the blot in the figure panel. The MYMX blots in the source data for Fig. 10A and F seem not to match the blots in the figure panels. The same is true for the total protein blot of Figure 10F. Please carefully check the composition of these panels and the corresponding source data and clarify.
 - Please describe your findings in the Abstract in present tense. Moreover, you note that it is "unexpeted" that TRIM28 functions independently of S473 phosphorylation. I think for the 'general' reader it is unclear why such a finding is unexpected and you might either want to add a sentence on the expected role of S473 or re-phrase the sentence.
 - The title reads somewhat unspcific. What about changing it to "Satellite cell-derived TRIM28 is essential for mechanical load- and injury-induced myogenesis"?
 - On a different note, I would like to alert you that EMBO Press offers a new format for a video-synopsis of work published with us, which essentially is a short, author-generated film explaining the core findings in hand drawings, and, as we believe, can be very useful to increase visibility of the work. This has proven to offer a nice opportunity for exposure i.p. for the first author(s) of the study. Please see the following link for representative examples and their integration into the article web page:
https://www.embopress.org/video_synopses
<https://www.embopress.org/doi/full/10.15252/emboj.2019103932>
- Please let me know, should you be interested to engage in commissioning a similar video synopsis for your work. According operation instructions are available and intuitive.
- We look forward to seeing a final version of your manuscript as soon as possible.

With kind regards,

Referee #2:

This reviewer requested the authors address four concerns. In the original comments #2 and #4, the authors fully addressed the concerns and added new data. This reviewer believes that the new data strongly supports the authors' conclusions.

In comment #1, the authors rebutted that the observation was a true hypertrophic response. If fusion was inhibited by TRIM28 deficiency, then an increase in the number of satellite cell-derived myonuclei should not occur, and consequently, the hypertrophic response should be inhibited. In 2-week SA model, Peterson group showed that the MW was no different in satellite cell-depleted mice (PMID: 21828094). However, in SA models of 8 weeks or longer, to the reviewer's knowledge, all literature

has indicated that the impaired functions of satellite cells attenuated hypertrophic responses. Therefore, this reviewer expected that the proposed experiments would show such a phenotype in TRIM28 cKO if no fusion occurs, reinforcing the conclusions. If the experiments were only in the hypertrophic model, this reviewer would request the authors to perform this experiment in this round. However, as this study showed that TRIM28 was involved in fusion even in the regeneration model, this reviewer will not mention this point any further.

In the comment #3, the authors did not add new data on the mutant-TRIM28. Some studies used the antibody from BioLegend (#654,102). Did the authors try this antibody? If the authors cannot use antibodies, they should show that the TRIM28 mutant generated here lacks phosphorylation-dependent function. Alternatively, the simplest way is to show evidence indicating that the authors succeeded in generating the mutated mice. In the M&Ms, it was not mentioned that the point mutation was confirmed by sequencing analysis, so, at least, it is necessary to confirm whether the theoretical construct was used in the experiments. Restriction enzyme results alone are not sufficient. Because there is a paper demonstrating the necessity of phosphorylation in myogenesis, the authors need to present the evidence.

Response to Reviewers

Referee #2:

This reviewer requested the authors address four concerns. In the original comments #2 and #4, the authors fully addressed the concerns and added new data. This reviewer believes that the new data strongly supports the authors' conclusions.

In comment #1, the authors rebutted that the observation was a true hypertrophic response. If fusion was inhibited by TRIM28 deficiency, then an increase in the number of satellite cell-derived myonuclei should not occur, and consequently, the hypertrophic response should be inhibited. In 2-week SA model, Peterson group showed that the MW was no different in satellite cell-depleted mice (PMID: 21828094). However, in SA models of 8 weeks or longer, to the reviewer's knowledge, all literature has indicated that the impaired functions of satellite cells attenuated hypertrophic responses. Therefore, this reviewer expected that the proposed experiments would show such a phenotype in TRIM28 cKO if no fusion occurs, reinforcing the conclusions. If the experiments were only in the hypertrophic model, this reviewer would request the authors to perform this experiment in this round. However, as this study showed that TRIM28 was involved in fusion even in the regeneration model, this reviewer will not mention this point any further.

We thank the reviewer for providing this insight. If we decide to further pursue this line of inquiry in the future, we will carefully examine this possibility.

In the comment #3, the authors did not add new data on the mutant-TRIM28. Some studies used the antibody from BioLegend (#654,102). Did the authors try this antibody? If the authors cannot use antibodies, they should show that the TRIM28 mutant generated here lacks phosphorylation-dependent function. Alternatively, the simplest way is to show evidence indicating that the authors succeeded in generating the mutated mice. In the M&Ms, it was not mentioned that the point mutation was confirmed by sequencing analysis, so, at least, it is necessary to confirm whether the theoretical construct was used in the experiments. Restriction enzyme results alone are not sufficient. Because there is a paper demonstrating the necessity of phosphorylation in myogenesis, the authors need to present the evidence.

Yes, we did try the BioLegend antibody that the reviewer mentioned and it did not work for us. We did however confirm the mutation using sequencing analysis. We have added the following statement to the Methods section, "The single point mutation of F0 and F1 founders was confirmed through amplicon sequencing followed with CRISPResso2 analysis."

Dr. Troy Hornberger
University of Wisconsin - Madison
Comparative Biosciences
2015 Linden Drive
Madison, WI 53706
United States

Dear Troy,

Thank you for updating the source data files. I am now very pleased to accept your manuscript for publication in the next available issue of EMBO reports. Thank you for your contribution to our journal.

Kind regards,

Martina

Corresponding Author Name: Troy Hornberger
Journal Submitted to: EMBO Reports
Manuscript Number: EMBOR-2024-58743V2

USEFUL LINKS FOR COMPLETING THIS FORM

- The EMBO Journal - Author Guidelines
- EMBO Reports - Author Guidelines
- Molecular Systems Biology - Author Guidelines
- EMBO Molecular Medicine - Author Guidelines